# Does Reinforcement Fine-Tuning Improve Generalization of LLM Agents? An Empirical Study

Zhiheng Xi [* 1]  Xin Guo [* 1]  Jiaqi Liu [1]  Jiazheng Zhang [1]  Yutao Fan [2]  Zhihao Zhang [1]  Shichun Liu [1]
Mingxu Chai [1]  Xiaowei Shi [3]  Yitao Zhai [3]  Xunliang Cai [3]  Tao Gui [1 4]  Qi Zhang [1]  Xuanjing Huang [1]

## Abstract

Reinforcement fine-tuning (RFT) has shown promise for training LLM agents to perform multi-turn decision-making based on environment feedback. However, most existing evaluations remain largely in-domain—training and testing are conducted in the same environment or even on the same tasks. In real-worlddeployment, agents may operate in unseen environments with different background knowledge, observation spaces, and action interfaces. To characterize the generalization profile of RFT under such shifts, we conduct a systematic study along three axes: (1) within-environment generalization across task difficulty, (2) cross-environment transfer to unseen environments, and (3) sequential multi-environment training to quantify transfer and forgetting. Our results show that RFT generalizes well across task difficulty within an environment, but exhibits weaker transfer to unseen environments, which correlates with shifts in both semantic priors and observation/action interfaces. In contrast, sequential training yields promising downstream gains with minimal upstream forgetting, and mixture training across environments improves the overall balance. We further provide detailed analyses and deeper insights, and hope our work helps the community develop and deploy generalizable LLM agents.

## 1. Introduction

Reinforcement fine-tuning (RFT) has emerged as a promising post-training paradigm for improving large language

*Equal contribution  [1]College of Computer Science and Artificial Intelligence, Fudan University, Shanghai, China [2]Shanghai Artificial Intelligence Laboratory, Shanghai, China [3]Meituan, Beijing, China [4]Shanghai Innovation Institute, Shanghai, China. Correspondence to: Tao Gui <tgui@fudan.edu.cn>.

*Proceedings of the $43^{rd}$ International Conference on Machine Learning*, Seoul, South Korea. PMLR 306, 2026. Copyright 2026 by the author(s).

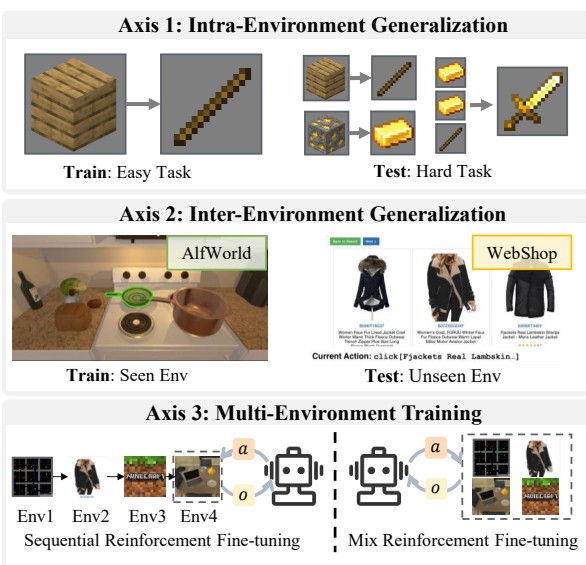

*Figure 1.* An overview of three axes we study.

model (LLM) agents on complex interactive tasks such as web navigation and software engineering (Zhou et al., 2023; Deng et al., 2023; He et al., 2024; Jimenez et al., 2023; Zan et al., 2025; Merrill et al., 2026). In RFT, an agent is trained to make a sequence of intelligent decisions that maximize task-specific objectives based on environment feedback (Xi et al., 2025c; Mai et al., 2025). By optimizing for long-horizon outcomes, RFT is often associated with stronger agentic behaviors such as instruction following (Ouyang et al., 2022a; Bai et al., 2022b), planning (Yao et al., 2022b; Shinn et al., 2023), reasoning (Lightman et al., 2023; Uesato et al., 2022), and tool use (Li et al., 2025b; 2023).

Despite rapid progress, most empirical evidence for RFT focuses on in-domain evaluation, where training and testing are conducted within the same environment—and often even on closely overlapping tasks (Zhou et al., 2023; Liu et al., 2023). In real-world deployment, however, agents frequently encounter unseen environments that differ not only in task instances, but also in background knowledge, observation spaces, and action interfaces (Ruan et al., 2023; Xi et al., 2025a). Such shifts can fundamentally reshape

the interaction dynamics faced by agents, including which observations are informative, what actions are feasible, and how failures can be corrected. This raises a practical and important research question: do the improvements brought by RFT generalize beyond the training distribution?

To bridge this gap, we conduct a systematic study to explore how RFT impacts the generalization and transferability of LLM agents across three axes (Figure 1). First, we investigate task-level generalization under the same environment by reviewing varying difficulty performance when RFT on a specific subset of tasks (Section 4); Second, we examine environment-level generalization to assess whether RFT maintains efficacy in unseen environments, where agents must navigate shifts in background knowledge alongside changes in observation and action spaces (Section 5). Third, we analyze sequential training across environments to characterize the dynamics of transfer and forgetting, comparing this approach against joint training on environmental mixtures (Section 6). Together, these three axes provide a comprehensive, systematic framework for understanding RFT-driven generalization and transfer in LLM agents.

Our analysis covers five agentic environments that vary in background and core properties, as shown in Table 1 and Table 4. We highlight several key findings: (1) Intra-environment generalization, within a consistent environment, RFT-trained agents exhibit significant generalization, and easy-to-hard curriculum learning further boosts performance gains. (2) Inter-environment sensitivity, while RFT enhances agentic capabilities, its generality to unseen environments exhibits fluctuations. Our analysis reveals that this across-environment generalization is sensitive to the required prior knowledge, observation spaces, and action spaces. (3) Multi-environment training. Employing multi-environment training strategies substantially enhances generalization: sequential RFT enables effective transfer to new environments while maintaining performance on upstream ones, and mix RFT across multiple environments achieves consistently strong results overall. We further provide insights that shed light on the mechanisms underlying these behaviors. Overall, we hope these findings inform the development of more generalizable LLM agents and their practical deployment in real-world settings.

## 2. Related Work

### 2.1. RL for LLM Agents

Reinforcement learning (RL) has been widely adopted in LLM training, with paradigms such as reinforcement learning from human feedback (RLHF) (Christiano et al., 2017; Ouyang et al., 2022b; Bai et al., 2022a) and reinforcement learning from verifiable rewards (RLVR) (Schulman et al., 2017; Rafailov et al., 2023; DeepSeek-AI, 2025) demon-

*Table 1.* Characteristics of each environment we study. The columns "DEN.", "VAL.", "ACT.", "KNWL." and "STR." indicate whether the rewards are *dense*, action validation is *strict*, *valid action lists* are provided per step, *world knowledge* is required, and observations are *structured*, respectively. The environments vary along these dimensions.

| Environment | Types | DEN. | VAL. | ACT. | KNWL. | STR. |
|---|---|---|---|---|---|---|
| WebShop | *Web* | ✔ | ✗ | ✗ | ✔ | ✔ |
| SearchQA | *Search* | ✗ | ✔ | ✗ | ✔ | ✔ |
| TextCraft | *Game* | ✗ | ✔ | ✗ | ✗ | ✗ |
| AlfWorld | *Household* | ✗ | ✔ | ✗ | ✗ | ✗ |
| BabyAI | *Embodied* | ✔ | ✗ | ✔ | ✗ | ✗ |

strating strong effectiveness in improving instruction following, reasoning accuracy, and behavioral alignment (Mu et al., 2024; Guo et al., 2025; Ren et al., 2025). Building upon these advances, recent work has extended reinforcement learning to LLM agents to enhance multi-step decision making (Zhai et al., 2024; 2025; Wang et al., 2025a), long-horizon planning (Song et al., 2024a; Xi et al., 2025c; Wang et al., 2025b), and interaction with external environments (Tan et al., 2024; Feng et al., 2024; Chen et al., 2025b). Across diverse agent settings, RL has been shown to strengthen agents' abilities to decompose complex tasks, coordinate reasoning with action , and adapt policies through environment feedback (Tian et al., 2024), enabling more effective information gathering (Zhang et al., 2025c; Ramrakhya et al., 2025; Jin et al., 2025), iterative self-correction (Kumar et al., 2025; Ma et al., 2025; Zeng et al., 2025a), and tool utilization (Zeng et al., 2025b; Feng et al., 2025; Li et al., 2025a). Despite these successes, most RL-based agent methods are evaluated primarily under in-domain settings with similar task distributions and interfaces, leaving open the question of how well such learned decision-making policies generalize across tasks or transfer to unseen environments.

### 2.2. Generalization and Forgetting by Post-training

Recent work has examined how post-training strategies affect the generalization and forgetting behaviors of LLMs (Zhao et al., 2024; Zhang et al., 2025d; Tu et al., 2025). Supervised fine-tuning (SFT) is effective for improving in-distribution performance but often leads to over-specialization and degradation of general capabilities due to representation and distributional drift (Kumar et al., 2022; Luo et al., 2023; Kotha et al., 2024; Wang et al., 2024). In contrast, RL, particularly RLVR, tends to better preserve pre-trained representations by optimizing trajectory-level objectives rather than directly fitting target distributions, enabling more robust transfer across tasks and domains (Huan et al., 2025; Chen et al., 2025a; Cheng et al., 2025; Chu et al., 2025). However, RL can also induce negative interference and winner-take-all dynamics, reducing behavioral cover-

age and leading to reasoning boundary shrinkage (Nguyen et al., 2025; Hu et al., 2025; Sun et al., 2025). Notably, most existing studies focus on static and single-turn LLM tasks, whereas we extend the study of post-training generalization and forgetting to multi-turn LLM agents and systematically evaluate these effects across different environments with distinct observation and action spaces.

# 3. Preliminaries

## 3.1. Preliminaries

**Task Formulation.** We follow the ReAct interaction paradigm (Yao et al., 2022b) to formulate a multi-turn decision-making task. The task can be represented by the tuple $(\mathcal{U}, \mathcal{S}, \mathcal{A}, \mathcal{O}, \mathcal{T}, \mathcal{R})$. Here, $\mathcal{U}$ denotes the instruction space, $\mathcal{S}$ is the state space, $\mathcal{A}$ is the action space, and $\mathcal{O}$ represents the observation space. The function $\mathcal{T} : \mathcal{S} \times \mathcal{A} \to \mathcal{S}$ denotes the deterministic state transition function, and $\mathcal{R} : \mathcal{U} \times \mathcal{S} \to \mathbb{R}$ is the reward function.

Given a task instruction $u \in \mathcal{U}$, the agent operates based on a policy $\pi_\theta$ (an LLM parameterized by $\theta$). In each interaction step $t$, the state $s_t$ consists all previous dialogues and their resulting observations, *i.e.*, $s_t = (a_0, o_0, \ldots, a_{t-1}, o_{t-1})$. The agent generates an action $a_t \sim \pi_\theta(\cdot \mid u, s_t)$. The action $a_t$ includes an internal reasoning trace and a interaction to the environment. The agent then receives an observation $o_k \in \mathcal{O}$ from the environment. This interaction loop continues until the tesk is completed or the maximum number of turns is reached, resulting in a complete trajectory:

$$\tau = \{u, (a_0, o_0), (a_1, o_1), \ldots, (a_T, o_T)\} \quad (1)$$

The environment provides a terminal reward $\mathcal{R}(\tau) \in [0, 1]$ upon task completion.

**Policy Gradient.** In RL, our objective is to optimize the policy parameters $\theta$ to maximize the expected cumulative reward over all possible trajectories for the given task (Sutton et al., 1998):

$$\max_\theta J(\theta) = \mathbb{E}_{\tau \sim \pi_\theta}[\mathcal{R}(\tau)] \quad (2)$$

To optimize the objective function $J(\theta)$, we utilize policy gradient methods (Sutton et al., 1999). Unlike value-based methods that approximate a value function, policy gradient methods directly search the policy parameter space. The core idea is to perform gradient ascent to update $\theta$, increasing the probability of trajectories that yield high rewards. The vanilla policy gradient is formulated as:

$$\nabla_\theta J(\theta) = \mathbb{E}_{\tau \sim \pi_\theta}\left[\mathcal{R}(\tau) \sum_{t=0}^{T} \nabla_\theta \log \pi_\theta(a_t|s_t)\right] \quad (3)$$

Based on this gradient estimation, the parameters are updated via $\theta_{new} = \theta_{old} + \alpha \nabla_\theta J(\theta)$, where $\alpha$ is the learning rate. However, standard policy gradient methods often suffer from high variance, leading to unstable training dynamics. To address this challenge, we adopt the widely-used GRPO algorithm (DeepSeek-AI, 2025), detailed in Appendix C.

## 3.2. Experimental Settings

**Environment Setup.** We select five representative agent environments, including WebShop (Yao et al., 2022a), SearchQA (Dunn et al., 2017), TextCraft (Sanghi et al., 2022), AlfWorld (Shridhar et al., 2020), and BabyAI (Chevalier-Boisvert et al., 2018), with characteristics of each environment summarized in Table 1. Among them, WebShop is an interactive web shopping environment; SearchQA is a Q&A environment augmented with context from a search engine; ALFWorld requires agents to explore rooms and execute tasks in a household setting; BabyAI is an interactive grid world simulator; and TextCraft is a game environment for crafting items in Minecraft. The task data $\mathcal{U}$ is sourced from AgentGym (Xi et al., 2025b), with detailed statistics and action spaces for each environment provided in Appendix B.

**Training Setup.** We perform RFT training using the AgentGym-RL framework (Xi et al., 2025c) with Qwen2.5-3B-Instruct and Qwen2.5-7B-Instruct (Team, 2024). Beyond GRPO (DeepSeek-AI, 2025), we further extend our evaluation to the REINFORCE++ (Hu, 2025) algorithm, with the results provided in Appendix F. In addition, we compare the performance of RFT with SFT (see Appendix G), highlighting the superiority of RFT in both in-domain and out-of-domain scenarios. Following the ReAct (Yao et al., 2022b) paradigm, each action receives real-time environment feedback. For all experiments, we sample 8 trajectories per task and set the maximum response length to 8192 tokens. Following Xi et al. (2025b), we set the maximum interaction turns $K$ for each environment as follows: 10 for WebShop, 5 for SearchQA, 30 for AlfWorld, 10 for BabyAI, and 15 for TextCraft.

**Evaluation Setup.** As summarized in Table 1, WebShop and BabyAI provide dense rewards, while SearchQA, AlfWorld, and TextCraft use binary rewards. We follow DeepSeek-AI (2025) and adopt exact matching, counting an action as correct only if it exactly matches the reference. To reduce randomness, we report `avg@8` as the main metric, and provide the confidence intervals in Appendix D to demonstrate the stability of our results. Besides, we also measure the average interaction turns ($\bar{k}$) and generated tokens ($\bar{l}$) for efficiency. For fair comparison, all agents are evaluated with the maximum number of interaction turns set to $K = 20$ during testing.

*Table 2.* Results of generalization across task difficulties within the same environment.

| Models | WebShop | | | SearchQA | | | TextCraft | | | AlfWorld | | | BabyAI | | |
|---|---|---|---|---|---|---|---|---|---|---|---|---|---|---|---|
| | *easy* | *hard* | *all* | *easy* | *hard* | *all* | *easy* | *hard* | *all* | *easy* | *hard* | *all* | *easy* | *hard* | *all* |
| *Qwen2.5-3B-Instruct* | | | | | | | | | | | | | | | |
| base model | 21.7 | 10.6 | 15.3 | 63.7 | 6.5 | 23.7 | 23.7 | 2.3 | 14.5 | 26.8 | 8.0 | 13.2 | 71.5 | 48.0 | 61.6 |
| train w/ $\mathcal{U}_{easy}$ | 90.3 | 75.3 | 81.6 | 82.7 | 16.9 | 36.6 | 97.6 | 49.4 | 76.9 | 93.2 | 82.0 | 85.1 | 93.4 | 79.0 | 87.3 |
| train w/ $\mathcal{U}_{hard}$ | 86.1 | 84.5 | 85.2 | 72.5 | 19.4 | 35.3 | 95.0 | 42.2 | 72.3 | 96.1 | 92.9 | 93.8 | 92.3 | 77.3 | 86.0 |
| train w/ $\mathcal{U}$ | 92.8 | 84.3 | 87.9 | 87.6 | 22.1 | 41.8 | 93.9 | 47.4 | 73.9 | 97.0 | 89.8 | 91.8 | 93.2 | 77.5 | 86.6 |
| $\mathcal{U}_{easy} + \mathcal{U}_{hard}$ | 93.6 | 85.2 | 88.7 | 82.1 | 19.8 | 38.5 | 93.9 | 52.9 | 76.3 | 97.3 | 93.4 | 94.4 | 94.5 | 82.2 | 89.3 |
| $\mathcal{U}_{hard} + \mathcal{U}_{easy}$ | 90.2 | 82.4 | 85.7 | 82.0 | 18.7 | 37.7 | 97.8 | 40.1 | 73.0 | 95.5 | 92.3 | 93.6 | 94.5 | 80.4 | 88.6 |
| *Qwen2.5-7B-Instruct* | | | | | | | | | | | | | | | |
| base model | 44.1 | 17.4 | 28.6 | 79.6 | 10.4 | 31.2 | 46.9 | 16.0 | 33.6 | 40.2 | 21.4 | 26.6 | 80.1 | 47.9 | 67.0 |
| train w/ $\mathcal{U}_{easy}$ | 88.1 | 77.5 | 82.0 | 85.8 | 21.1 | 40.5 | 98.2 | 47.4 | 76.4 | 95.2 | 89.7 | 91.2 | 95.2 | 76.1 | 87.2 |
| train w/ $\mathcal{U}_{hard}$ | 87.9 | 81.4 | 84.2 | 93.3 | 27.0 | 46.9 | 99.3 | 57.3 | 81.3 | 97.5 | 94.6 | 95.4 | 92.8 | 77.9 | 86.5 |
| train w/ $\mathcal{U}$ | 92.9 | 81.9 | 86.5 | 91.8 | 26.6 | 46.1 | 95.2 | 52.3 | 80.9 | 97.7 | 97.2 | 92.0 | 95.6 | 74.5 | 88.8 |
| $\mathcal{U}_{easy} + \mathcal{U}_{hard}$ | 90.6 | 77.5 | 83.0 | 89.8 | 25.5 | 44.8 | 98.7 | 64.0 | 83.8 | 96.1 | 92.6 | 93.6 | 95.5 | 82.9 | 90.2 |
| $\mathcal{U}_{hard} + \mathcal{U}_{easy}$ | 89.1 | 79.1 | 83.3 | 87.6 | 21.5 | 42.3 | 99.8 | 57.0 | 81.4 | 97.7 | 88.5 | 91.1 | 92.5 | 78.2 | 86.5 |

# 4. Does RFT Yield Generalization Across Task Difficulties within the Same Environment?

Reinforcement fine-tuning trains LLM agents through interaction with the environment, allowing them to learn its dynamics and adapt over time (Song et al., 2024b; Cui et al., 2025). Yet even with a fixed action and observation space, tasks within the same environment can differ substantially in exploration depth and information accessibility. In this section, we investigate *whether RFT policies learned on a subset of tasks transfer to other tasks of differing difficulty within the same environment.*

**Setting.** Following practice in previous work (Bengio et al., 2009; Mukherjee et al., 2023), we categorize the tasks $\mathcal{U}$ into *easy* and *hard* difficulty levels (denoted as $\mathcal{U}_{easy}$ and $\mathcal{U}_{hard}$) based on the avg@8 results of the Qwen2.5-7B-Instruct model, while ensuring a balanced distribution of data between the two difficulty levels. The same categorization is applied to the test set. Detailed data statistics for each environment are provided in Appendix B.

**RFT demonstrates strong tranferability across varying difficulty levels within the same environment.** As shown in Table 2, RFT demonstrates strong robustness when trained on data of varying difficulty levels within the same environment. Notably, with 7B model on WebShop, training on $\mathcal{U}_{easy}$ improves performance on the *hard* testset by 60.1 points, suggesting that RFT encourages agents to adapt to the environment, thereby enabling performance transfer across tasks with different difficulty levels and varying steps (Huan et al., 2025; Chen et al., 2025a). Overall, we also find that training on $\mathcal{U}_{hard}$ yields more gains on test set, e.g., for 3B model in AlfWorld, training on $\mathcal{U}_{hard}$ outperforms

training on $\mathcal{U}_{easy}$ by 8.7 points. We attribute this trend to the richer failure signals and longer-horizon exploration induced by harder tasks, which push the RFT optimizer to seek better policies (Xu et al., 2023; Wang et al., 2023).

**Curriculum learning can further enhance performance.** Additionally, we investigate the impact of the training sequence using data of varying difficulty levels. Results in Table 2 indicate that training on $\mathcal{U}$ (i.e., mixture of $\mathcal{U}_{easy}$ and $\mathcal{U}_{hard}$) generally does not yield optimal performance. In most cases, training on $\mathcal{U}_{easy}$ first, followed by $\mathcal{U}_{hard}$, achieves the best results. Compared with using a single difficulty level, this easy-to-hard curriculum enables further performance improvement. For example, on BabyAI, training on $\mathcal{U}_{easy} + \mathcal{U}_{hard}$ outperforms training on $\mathcal{U}_{easy}$ and on $\mathcal{U}_{hard}$ by 2.0 and 3.3 points, respectively. This result validates the rationale and effectiveness of curriculum learning (Qi et al., 2024; Zhang et al., 2025b).

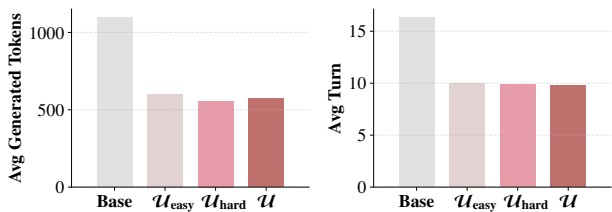

*Figure 2.* Average generated tokens and interactive turn of all environments for Qwen2.5-3B-Instruct model trained with varying-difficulty tasks.

**RFT training encourages agents to perform efficient exploration.** Figure 2 reports the average number of interaction turns and generated tokens across different environments, with a more detailed breakdown provided in

*Table 3.* Results of generalization across different environments. Darker red indicates greater performance improvement, while darker blue signifies more pronounced performance decline. The best result is in **bold**, while the second-best is marked with underline.

| Models | WebShop | SearchQA | TextCraft | AlfWorld | BabyAI | ΔHeld-In | ΔHeld-Out | ΔOverall |
|---|---|---|---|---|---|---|---|---|
| *Qwen2.5-3B-Instruct* | | | | | | | | |
| base model | 15.30 | 23.66 | 14.50 | 13.19 | 61.83 | − | − | − |
| train w/ WebShop | 87.86 | 25.84 | 18.50 | 23.06 | 70.91 | +72.56 | **+6.29** | +19.54 |
| train w/ SearchQA | 22.97 | 41.78 | 22.75 | 12.13 | 65.72 | +18.13 | +4.69 | + 4.56 |
| train w/ TextCraft | 14.46 | 22.16 | 73.88 | 11.00 | 63.73 | +59.38 | −0.66 | +11.35 |
| train w/ AlfWorld | 21.86 | 23.50 | 24.88 | 91.81 | 64.68 | **+78.62** | +4.91 | **+19.65** |
| train w/ BabyAI | 28.36 | 22.63 | 16.75 | 4.50 | 86.55 | +24.72 | +1.40 | + 6.06 |
| *Qwen2.5-7B-Instruct* | | | | | | | | |
| base model | 28.59 | 31.19 | 33.63 | 26.56 | 67.00 | − | − | − |
| train w/ WebShop | 86.50 | 33.28 | 40.75 | 24.13 | 79.21 | +57.91 | +4.75 | +15.38 |
| train w/ SearchQA | 47.07 | 46.12 | 35.25 | 16.75 | 80.33 | +14.93 | +5.91 | + 7.71 |
| train w/ TextCraft | 38.30 | 32.19 | 80.88 | 31.50 | 77.95 | +47.25 | **+6.65** | +14.77 |
| train w/ AlfWorld | 34.31 | 29.59 | 36.13 | 92.00 | 72.91 | **+65.44** | +3.13 | **+15.59** |
| train w/ BabyAI | 10.25 | 29.41 | 39.25 | 28.13 | 88.79 | +21.79 | −3.23 | + 1.77 |

Figure 8 of Appendix I. We find that RFT enables agents to interact and explore more efficiently within the same environment. For example, when training the 3B model on $\mathcal{U}_{\text{easy}}$ in BabyAI, the average interaction turns decrease from 10.76 to 4.19, and the average trajectory length is reduced from 624.58 to 160.60 tokens. These results indicate that RFT not only improves success rates but also encourages more concise and goal-directed exploration, substantially enhancing interaction efficiency (Nakano et al., 2021; Song et al., 2024b). A case can be found in Appendix K.

## 5. Does RFT Yield Generalization Across Different Environments?

In real-world scenarios, agents may encounter previously unseen environments and tasks, which is essential for practical deployment. We therefore investigate *whether RFT improves performance in unseen environments with different background knowledge, observation space and action spaces*. Concretely, we perform RFT training within a single environment and then evaluate the trained agent on other environments to measure cross-environment generalization.

**Setting.** We employ the base model (i.e., Qwen2.5-3B-Instruct or Qwen2.5-7B-Instruct) as the baseline and evaluate its `avg@8` metric across varying environments. To make these comparisons explicit, we define three metrics: ΔHeld-In, ΔHeld-Out, and ΔOverall. ΔHeld-In measures the average improvement over the baseline when the training and test environments coincide. ΔHeld-Out measures the average improvement when the agent is evaluated on environments different from those seen during training. Finally, ΔOverall reports the average improvement over the baseline across all evaluation settings.

**Performance differs significantly between held-in and held-out environments.** Table 3 reveals that agents exhibit generalization across different environments, yet a significant performance gap exists between held-in and held-out settings. Specifically, substantial gains are achieved under held-in conditions. In AlfWorld, performance improves by 78.62 and 65.44 points for the 3B and 7B models, respectively. In contrast, in held-out conditions, generalization remains possible in most cases, but with more modest gains. On average, the 3B and 7B models yield improvements of 3.32 and 3.44 points on unseen environments, respectively.

**In unseen environments, positive transfer is observed in most cases.** Compared to the baseline, most trained agents can demonstrate transferability to unseen environments, even when these environments demand different background knowledge and operate under different action spaces. Notably, agents trained on WebShop, AlfWorld, and SearchQA consistently exhibit positive Δheld-out. When evaluated on WebShop, agents trained on SearchQA yield performance gains of 7.67 and 18.48 points for the 3B and 7B models, respectively. This can be attributed to the similarity between WebShop and SearchQA as search-based environments. Specifically, the agent trained on SearchQA learns to formulate more flexible search queries, as well as efficient information extraction from complex results. An illustrative case is presented in Figure 10 in Appendix K and discussed in Section 7.

**In unseen environments, negative effects may sometimes occur.** However, agents trained on TextCraft and BabyAI may struggle to generalize to other environments, e.g., after training on BabyAI, the 7B models show average negative improvements of –3.23 on held-out tasks, and even drop sharply from 28.59 to 10.25 on WebShop. Through care-

ful analysis, we find that because the BabyAI environment provides available actions at each step, the agent gradually becomes dependent on this information during training, leading to a decline in long-horizon reasoning capability. When faced with other environments, it fails to accurately use the valid action, resulting in a sharp performance drop. A case of interaction between the agent and BabyAI environment is provided in Appendix K.

**Cross-environment generalization performance varies significantly across target environments.** Moreover, our observations reveal that cross-environment generalization performance is highly dependent on the target environment. On the one hand, agents trained on nearly all source environments generalize effectively to TextCraft and BabyAI, as these domains rely less on specific background knowledge, thereby allowing acquired skills to transfer more readily. On the other hand, effective transfer to AlfWorld and SearchQA proves to be significantly more challenging. We attribute this to the strict action validation and sparse feedback inherent in these two environments. For instance, AlfWorld responds uniformly with "Nothing happens." to all invalid actions, offering no instructional guidance for improvement.

## 6. How Does Sequential RFT Across Environments Affect Transfer and Forgetting?

Beyond zero-shot transfer to unseen environments without additional training, we further investigate the effect of sequential training across multiple environments on agentic behaviors, specifically the resulting learning dynamics on memorizing downstream tasks or forgetting on upstream ones. In our experimental setup, a model initially converged in one environment is further trained on a second environment, after which we assess both its retention of performance on the original environment and its adaptation to

the new one. Finally, we extend this analysis to sequential training across five environments, comparing this strategy with joint training across a mixture of environments.

**Setting.** We conduct 20 two-stage experiments by sequentially pairing each of the five environments as the upstream and the downstream, as well as several five-stage training experiments. Using agents trained on single-environment as baselines, we evaluate anti-forgetting by upstream performance retention and transferability by downstream performance gains. Furthermore, we compare the five-stage sequential approach with joint training, in which the agent is trained on all available data from every environment in a randomly mixed manner.

**Sequential training demonstrates consistent anti-forgetting and transferability.** Figure 3 illustrates the performance dynamics of 8 two-stage sequential training scenarios, with others and the final results summarized in Figure 7 and Table 11 in Appendix H. Overall, sequential training matches or exceeds single-task performance on the downstream environment while largely preserving upstream performance. For instance, when the WebShop-pre-trained agent is further trained on TextCraft, it boosts performance on TextCraft from the single-task baseline of 80.88 to 82.50. Meanwhile, performance on WebShop experiences only a minor fluctuation, shifting from 86.5 to 86.32. These findings suggest that sequential training endows agents with stable capabilities for transfer and resistance to forgetting.

**Generalization in multi-environment training is highly correlated with single-environment training.** Figure 4 demonstrates the generalization performance of multi-environment training, revealing a strong alignment with the generalization patterns in single-environment training. First, environments that yield poor generalization in single-environment settings continue to be detrimental. For in-

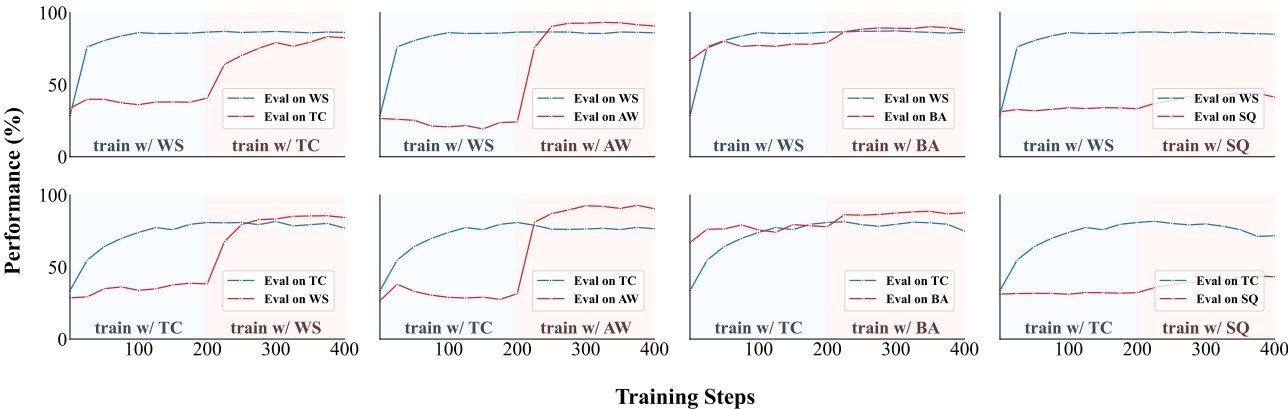

*Figure 3.* Training dynamics of forgetting and transfer in sequential two-stage cross-environment training with Qwen2.5-7B-Instruct, where blue and red denote the upstream environment and the downstream environment, respectively.

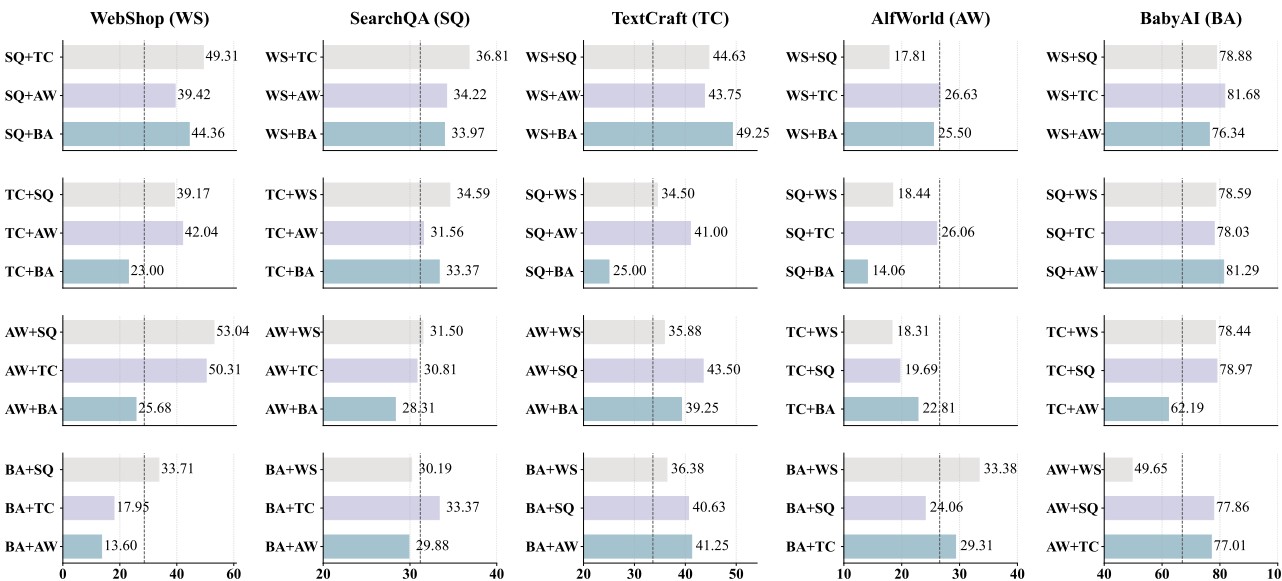

*Figure 4.* Generalization of sequential two-stage cross-environment training with Qwen2.5-7B-Instruct. Across five environments (WS, SQ, TC, AW, BA), the figure presents the generalization performance on the *three unseen environments* following sequential training on two environments. Each subplot corresponds to a fixed first environment, with *dashed line* indicating the baseline performance.

stance, the agent trained on BabyAI exhibits severe negative effect on WebShop. Even when BabyAI is trained sequentially after other generalizable environments (i.e., TC+BA, AW+BA), it drastically degrades their WebShop performance. Notably, the score of TC+BA dropped sharply from 38.30 (score of TC in Table 3) to 23.0.

Moreover, from the perspective of target environments, the results also show consistency with single-environment generalization. For target environments easy to generalization in single-environment scenarios, such as TextCraft and BabyAI, sequentially trained agents also tend to perform well. Specifically, agents pre-trained on WebShop (i.e., WS+SQ, WS+AW, WS+BA) achieve significant gains on TextCraft, increasing by 11.00, 10.12, and 15.62 points, respectively. However, for environments challenging for generalization, like AlfWorld, sequential training yields limited benefits. For instance, all three agents utilizing TextCraft as the upstream environment (TC+WS, TC+SQ, TC+BA) suffer performance degradation on AlfWorld.

**Training order significantly affects generalization performance in held-out environments.** As shown in Figure 4, the training order exerts a substantial influence on generalization performance. For instance, when evaluated on TextCraft and AlfWorld, BA+SQ outperforms SQ+BA by 15.63 and 10.00 points, respectively. This represents a substantial margin, particularly for held-out scenarios. We attribute this phenomenon to the inherent task difficulty within each environment. As noted in Section 5, BabyAI provides detailed feedback, whereas SearchQA imposes strict con-

strains with limited feedback. Consequently, the BA+SQ order naturally creates an "easy-to-hard" curriculum, which in turn facilitates better generalization performance.

**Sequential training achieves performance comparable to joint training.** Furthermore, we conduct sequential training over five environments in different orders. Figure 5 illustrates the performance dynamics on each environment across the five training stages for two representative sequences; for comparison, joint-training results are indicated by dashed lines. The results show that sequential training achieves performance comparable to joint training, even after training on five distinct tasks. Overall, the final performance is insensitive to the training order, which we attribute to RFT's ability to preserve previously acquired capabilities, consistent with findings in prior work. For environments such as AlfWorld and SearchQA—to which other tasks struggle to generalize—relatively pronounced forgetting may occur over the course of long-term sequential training.

## 7. Further Analysis and Discussion

**Failure Mode Analysis.** We conduct a fine-grained error analysis on both intra/inter-environment evaluations, summarizing 8 common failure modes including *instruction misinterpretation*, *action execution failure*, and *logical deficits*. Using GPT-5-mini, we systematically categorize the error trajectories across all scenarios. More detail about each failure pattern can be found in Appendix J.

The results presented in Figure 6 reveal that while failure

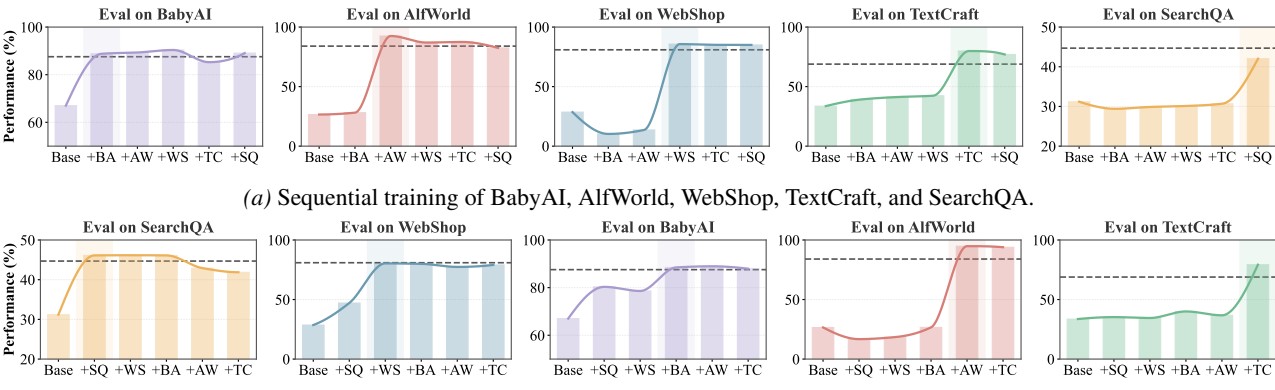

*(a)* Sequential training of BabyAI, AlfWorld, WebShop, TextCraft, and SearchQA.

*(b)* Sequential training of SearchQA, WebShop, BabyAI, AlfWorld, and TextCraft.

*Figure 5.* Training dynamics of sequential training across five environments. We present the results for representative sequence combinations, monitoring how performance on each environment changes as the agent is trained on different environments sequentially. The *dashed lines* denote the performance achieved by **joint training on a mixture of data from all five environments**.

modes differ by environment, errors related to "Confirmation Bias" are prevalent ($> 10\%$) across all scenarios. This suggests that after training, agents tend to exhibit overconfidence, neglecting further verification and lacking the capacity for self-reflection based on environmental feedback. Moreover, in SearchQA, errors categorized as "Guessing or Fabrication" are widespread among both held-in ($23.8\%$) and held-out ($21.2\%$) settings. This highlights a critical deficiency in tool utilization, which fundamentally constrains the agent's generalization potential.

When the training and test environments differ, agents exhibit different failure modes. For example, in WebShop we observe an increase in the proportion of "State or Memory Inconsistency" errors, rising from $4.3\%$ in held-in to $21.9\%$ in held-out settings. This suggests that, when confronted with large volumes of information, the agent's decision-making becomes less coherent, and it struggles to generalize its ability to extract salient information. A full environment-by-environment breakdown is provided in Figure 9.

**Case Study.** We perform a comprehensive analysis of environment generalization and present specific case studies in Appendix K. Notably, Figure 10 illustrates the mechanism behind the successful transfer from SearchQA to WebShop. Specifically, the base agent tends to blindly input the entire instruction into the search bar, and struggles to accurately extract key information from the voluminous HTML content, resulting in the retrieval and selection of irrelevant items. In contrast, the agent trained on SearchQA learns to effectively search for key details and extract information, successfully selecting the correct item.

Furthermore, Figure 11 offers qualitative insight into why SearchQA generalization remains difficult, using an AlfWorld-trained agent as a case study. Although both agents fail initially, the SearchQA-trained agent iteratively

refines its queries to become more specific, improving retrieval and ultimately recovering. By contrast, this query-refinement behavior does not reliably transfer: the AlfWorld-trained agent falls into a degenerate loop, repeatedly issuing near-duplicate searches, returning the same responses, and ultimately failing.

## 8. Conclusion

In this paper, we present a systematic study of how RFT affects the transfer and generalization of LLM agents for multi-turn decision-making. Through large-scale experiments along three complementary axes, we characterize when RFT generalizes within and across environments and identify generalization patterns. We further complement our quantitative results with failure mode analysis and qualitative case study to pinpoint where agents break down and what behaviors fail to transfer. Together, our findings offer practical guidance for training and evaluating agents under distribution shift, and we hope they inform the development of agents that generalize reliably in real-world deployments.

## Acknowledgements

The authors wish to thank the anonymous reviewers for their helpful comments. This work was partially funded by National Natural Science Foundation of China (No.62576106, 62476061, 62376061).

## Impact Statement

This paper studies the generalization and transfer behavior of reinforcement fine-tuning (RFT) for large language model (LLM) agents. In realistic deployments, agents face distribution shifts arising from changes in background knowledge, observation spaces, and action interfaces. As LLM agents

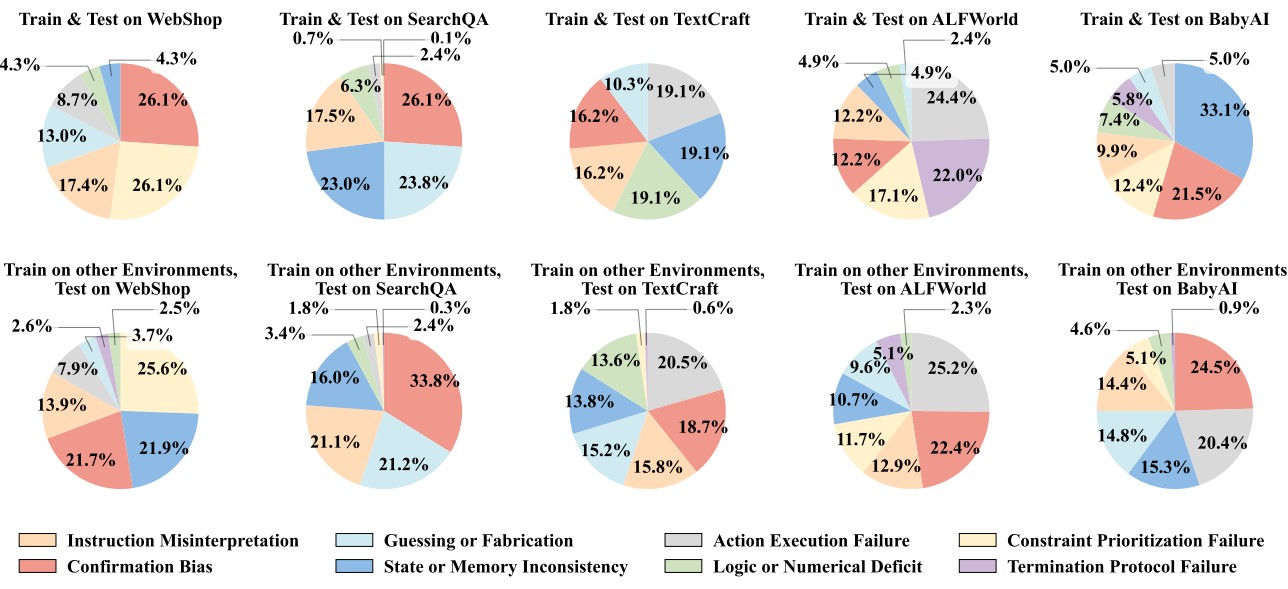

*Figure 6.* Comparison of failure mode distributions between in-environment and out-of-environment evaluation.

are increasingly used for long-horizon decision-making, it is crucial to assess whether gains achieved during training persist beyond the training distribution. We therefore conduct a systematic study across multiple representative environments along three complementary axes, yielding empirical findings and practical insights. Our experiments also highlight safety and reliability concerns. In particular, because RFT can boost downstream performance while retaining capabilities from upstream training, it may also preserve undesirable or unsafe behaviors learned earlier, creating latent risks (e.g., backdoor-like behaviors (Rando & Tramèr, 2023; Hubinger et al., 2024) that remain after fine-tuning). These findings underscore the need for stronger out-of-distribution evaluation and safety-aware training and auditing procedures for RFT-trained agents.

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

## A. Limitations and Future Work

This paper adopts a new perspective to investigate how reinforcement fine-tuning affects the generalization ability of LLM agents. Our experiments focus on the Qwen2.5 family and yield extensive empirical results together with substantive insights. Despite this progress, important challenges remain. First, we rely on default training/evaluation protocols and hyperparameter choices, without exhaustive tuning; more careful optimization may bring new findings. Second, we adopt GRPO—the most commonly used RL algorithm for LLM agents at present—rather than other approaches such as REINFORCE++ (Hu, 2025) or DAPO (Yu et al., 2025); we leave these for future work. Additionally, due to computational constraints, our sequential multi-environment training study does not enumerate all possible environment orderings; instead, we evaluate a small set of representative sequence. Although these experiments already reveal important insights, a broader and more fine-grained investigation of ordering effects is left for future work and may further substantiate our findings.

## B. Dataset Details

We select five representative agent environments, including WebShop (Yao et al., 2022a), SearchQA (Dunn et al., 2017), TextCraft (Sanghi et al., 2022), AlfWorld (Shridhar et al., 2020), and BabyAI (Chevalier-Boisvert et al., 2018). The characteristics of each environment are shown in Table 1 in Section 3.2. Here, we provide their detailed types and action spaces in Table 4.

*Table 4.* Detailed action spaces for each environment.

| Environment | Types | Action Spaces |
|---|---|---|
| WebShop | *Web Navigation* | *search, click* |
| SearchQA | *Q&A Search* | *search, answer* |
| TextCraft | *Text-based Game* | *get, craft, inventory* |
| AlfWorld | *Household* | *go to, open, close, take from, put in/on, use, heat, cool, clean, slice, inventory, look, examine* |
| BabyAI | *Embodied* | *turn right, turn left, move forward, go to, go through, toggle and go through, toggle, pickup, drop, check available actions* |

Our data is sourced from AgentGym (Xi et al., 2025b). Following practice in previous work (Bengio et al., 2009; Mukherjee et al., 2023), we categorize the tasks $\mathcal{U}$ into *easy* and *hard* difficulty levels (denoted as $\mathcal{U}_{\text{easy}}$ and $\mathcal{U}_{\text{hard}}$) based on the `avg@8` results of the Qwen2.5-7B-Instruct model, while ensuring a balanced distribution of data between the two difficulty levels. The same categorization is applied to the test set. Detailed statistics for each environment are provided in Table 5.

*Table 5.* Detailed data statistics for each environment.

| Environment | Training dataset | | | Testing dataset | | |
|---|---|---|---|---|---|---|
| | *easy* | *hard* | *all* | *easy* | *hard* | *all* |
| WebShop | 2104 | 1826 | 3930 | 84 | 116 | 200 |
| SearchQA | 1960 | 2040 | 4000 | 120 | 280 | 400 |
| TextCraft | 235 | 209 | 444 | 57 | 43 | 100 |
| AlfWorld | 1267 | 1153 | 2420 | 55 | 145 | 200 |
| BabyAI | 398 | 412 | 810 | 52 | 38 | 90 |

## C. Detailed Algorithm of GRPO

Group Relative Policy Optimization (GRPO) is an efficient online reinforcement learning algorithm tailored for LLMs. It eliminates the need for a separate critic network typically required in PPO, thereby reducing computational overhead. Instead, GRPO estimates the baseline using *group relative advantages* to significantly reduce gradient variance.

Specifically, for each input query $q$ (derived from $u$), GRPO samples a group of outputs $\{y_1, y_2, \ldots, y_G\}$ from the old policy $\pi_{\theta_{old}}$ and obtains their corresponding rewards $\{R_1, R_2, \ldots, R_G\}$. The advantage $A_i$ for each output is calculated by normalizing the rewards within the group:

$$A_i = \frac{r_i - \text{mean}(\{R_1, \ldots, R_G\})}{\text{std}(\{R_1, \ldots, R_G\})} \qquad (4)$$

Finally, GRPO updates the policy by maximizing the following surrogate objective, which incorporates PPO-style clipping (Schulman et al., 2017) and a KL divergence penalty to ensure training stability:

$$
\mathcal{J}_{\text{GRPO}}(\theta) = \mathbb{E} \Bigg[ \frac{1}{G} \sum_{i=1}^{G} \Big( \min\big(r_i(\theta) A_i,
$$

$$
\text{clip}\big(r_i(\theta), 1 - \epsilon, 1 + \epsilon\big) A_i\big) - \beta \mathbb{D}_{\text{KL}} \Big) \Bigg],
$$

(5)

where $r_i(\theta) = \frac{\pi_\theta(y_i|q)}{\pi_{\theta_{old}}(y_i|q)}$ denotes the probability ratio between the new and old policies, $\{y_i\}_{i=1}^{G} \sim \pi_{\theta_{old}}(q)$ represents the outputs $y_i$ sampled from the old policy $\pi_{\theta_{old}}$ given the query $q$, $\epsilon$ is the clipping parameter, and $\beta$ is the coefficient for the KL divergence term.

## D. Stability of Our Results

Taking the generalization results of Qwen2.5-3B-Instruct across different environments as an example, we report the confidence intervals in Table 6, demonstrating the stability of our conclusions.

Table 6. Confidence intervals for results of generalization across different environments.

| Models | WebShop | SearchQA | TextCraft | AlfWorld | BabyAI |
|---|---|---|---|---|---|
| train w/ WebShop | $87.86 \pm 0.18$ | $25.84 \pm 0.47$ | $18.50 \pm 1.50$ | $23.06 \pm 0.96$ | $70.91 \pm 0.48$ |
| train w/ SearchQA | $22.97 \pm 1.23$ | $41.78 \pm 0.20$ | $22.75 \pm 1.50$ | $12.13 \pm 0.25$ | $65.72 \pm 2.81$ |
| train w/ TextCraft | $14.46 \pm 0.26$ | $22.16 \pm 0.38$ | $73.88 \pm 1.00$ | $11.00 \pm 0.63$ | $63.73 \pm 1.04$ |
| train w/ AlfWorld | $21.86 \pm 0.35$ | $23.50 \pm 1.05$ | $24.88 \pm 2.06$ | $91.81 \pm 0.41$ | $64.68 \pm 1.59$ |
| train w/ BabyAI | $28.36 \pm 0.65$ | $22.63 \pm 0.83$ | $16.75 \pm 0.96$ | $4.50 \pm 0.41$ | $86.55 \pm 1.03$ |

## E. Ablation Study

We conduct ablation experiments on Qwen2.5-7B-Instruct to study the effects of the *learning rate* and *rollout_n* (set to $1e-6$ and $8$ in main experiments), and also evaluate stopping at different training steps (set to 200 in main experiments). The results in Table 7 show that different hyperparameter settings have only a small effect. Although more sufficient training and more careful hyperparameter tuning may lead to better performance, they do not affect our overall conclusions.

Table 7. Results under varying *learning rate* (1e-6, 2e-6 and 5e-7), *rollout_n* (8 and 4), and *stopping steps* (100, 150 and 200).

| Models | learning rate | rollout_n | stopping steps | WebShop | SearchQA | TextCraft | AlfWorld | BabyAI | Avg. |
|---|---|---|---|---|---|---|---|---|---|
| train w/ WebShop | $1 \times 10^{-6}$ | 8 | 200 | 86.50 | 33.28 | 40.75 | 24.13 | 79.21 | 52.8 |
| | $2 \times 10^{-6}$ | 8 | 200 | 86.09 | 30.50 | 36.00 | 25.41 | 78.17 | 51.2 |
| | $5 \times 10^{-7}$ | 8 | 200 | 82.86 | 33.66 | 33.50 | 25.94 | 76.28 | 50.5 |
| | $1 \times 10^{-6}$ | 4 | 200 | 84.03 | 31.87 | 37.50 | 24.56 | 78.69 | 51.3 |
| | $1 \times 10^{-6}$ | 8 | 150 | 85.59 | 34.03 | 38.00 | 23.75 | 78.18 | 51.9 |
| | $1 \times 10^{-6}$ | 8 | 100 | 86.13 | 34.00 | 36.13 | 23.63 | 77.24 | 51.4 |
| train w/ SearchQA | $1 \times 10^{-6}$ | 8 | 200 | 47.07 | 46.12 | 35.25 | 16.75 | 80.33 | 45.1 |
| | $2 \times 10^{-6}$ | 8 | 200 | 44.31 | 40.10 | 29.88 | 20.38 | 77.70 | 42.5 |
| | $5 \times 10^{-7}$ | 8 | 200 | 42.51 | 40.22 | 35.38 | 23.25 | 75.70 | 43.4 |
| | $1 \times 10^{-6}$ | 4 | 200 | 48.53 | 42.19 | 33.50 | 17.12 | 78.25 | 43.9 |
| | $1 \times 10^{-6}$ | 8 | 150 | 44.99 | 44.00 | 37.63 | 20.75 | 80.02 | 45.5 |
| | $1 \times 10^{-6}$ | 8 | 100 | 41.65 | 38.97 | 36.62 | 23.31 | 78.15 | 43.7 |

## F. Experiments of More Algorithms

To evaluate the generalization across different algorithms, we conduct experiments using REINFORCE++ (Hu, 2025). The results of cross-environment generalization, presented in Table 8, indicate that REINFORCE++ improves performance in the

held-in environment. In terms of cross-environment generalization, REINFORCE++ still demonstrates a moderate transfer capability, although its performance remains inferior to that of GRPO. These findings are consistent with prior studies (Xi et al., 2025c; Zhang et al., 2025a; Hu et al., 2025), and they also constitute one of the main reasons why we chose GRPO.

*Table 8.* Comparison of cross-environment generalization performance between REINFORCE++ and GRPO.

| Models | Method | TextCraft | AlfWorld | BabyAI |
|---|---|---|---|---|
| base model | - | 33.63 | 26.56 | 67.00 |
| train w/ TextCraft | REINFORCE++ | 48.25 | 22.63 | 71.04 |
| | GRPO | 80.88 | 31.50 | 77.95 |
| train w/ AlfWorld | REINFORCE++ | 34.88 | 45.44 | 69.86 |
| | GRPO | 36.13 | 92.00 | 72.91 |
| train w/ BabyAI | REINFORCE++ | 36.25 | 21.44 | 73.68 |
| | GRPO | 39.25 | 28.13 | 88.79 |

In addition, we validate the REINFORCE++ algorithm in the BabyAI environment. The results in Table 9 show that REINFORCE++ exhibits patterns consistent with GRPO: it demonstrates strong transferability across varying difficulty levels, and the integration of curriculum learning further improves its performance.

*Table 9.* Comparison of cross-task generalization performance in BabyAI environment between REINFORCE++ and GRPO.

| Models | *easy* | *hard* | *all* |
|---|---|---|---|
| base model | 80.96 | 47.90 | 67.00 |
| train w/ $\mathcal{U}_{easy}$ | 89.04 | 60.17 | 76.85 |
| train w/ $\mathcal{U}_{hard}$ | 91.44 | 63.03 | 79.44 |
| $\mathcal{U}_{easy} + \mathcal{U}_{hard}$ | 93.84 | 69.44 | 83.54 |

## G. Comparison with More Post-training Strategies

To compare with other post-training strategies, we conduct experiments on Qwen2.5-7B-Instruct using SFT. Results in Table 10 indicate that GRPO outperforms SFT on held-in environments and exhibits better generalization performance on held-out environments, further validating the advantages of RFT.

*Table 10.* Comparison of cross-environment generalization performance between RFT and SFT.

| Models | Method | WebShop | SearchQA | TextCraft | AlfWorld | BabyAI | $\Delta$Held-In | $\Delta$Held-Out |
|---|---|---|---|---|---|---|---|---|
| base model | - | 28.59 | 31.19 | 33.63 | 26.56 | 67.00 | − | − |
| train w/ WebShop | SFT | 77.46 | 23.63 | 10.13 | 1.00 | 32.51 | +48.87 | −22.78 |
| | GRPO | 86.50 | 33.28 | 40.75 | 24.13 | 79.21 | +57.91 | + 4.75 |
| train w/ TextCraft | SFT | 10.99 | 23.84 | 70.25 | 2.69 | 62.67 | +36.62 | −13.29 |
| | GRPO | 38.30 | 32.19 | 80.88 | 31.50 | 77.95 | +47.25 | + 6.65 |

## H. Detailed Results of Sequential Cross-Environment Training

Figure 3 in Section 6 illustrates the training dynamics for 8 two-stage sequential training configurations. Here, we present the dynamics for the remaining 12 pairs in Figure 7. Additionally, detailed final results are reported in Table 11.

## I. Detailed Results of Average Turns and Generated Tokens in Different Environment

Figure 2 in section 4 reports the average number of interaction turns and generated tokens across different environments. The more detailed results are reported in Figure 8.

## J. Failure Mode Analysis

Through detailed analysis, we summarize 8 common error types:

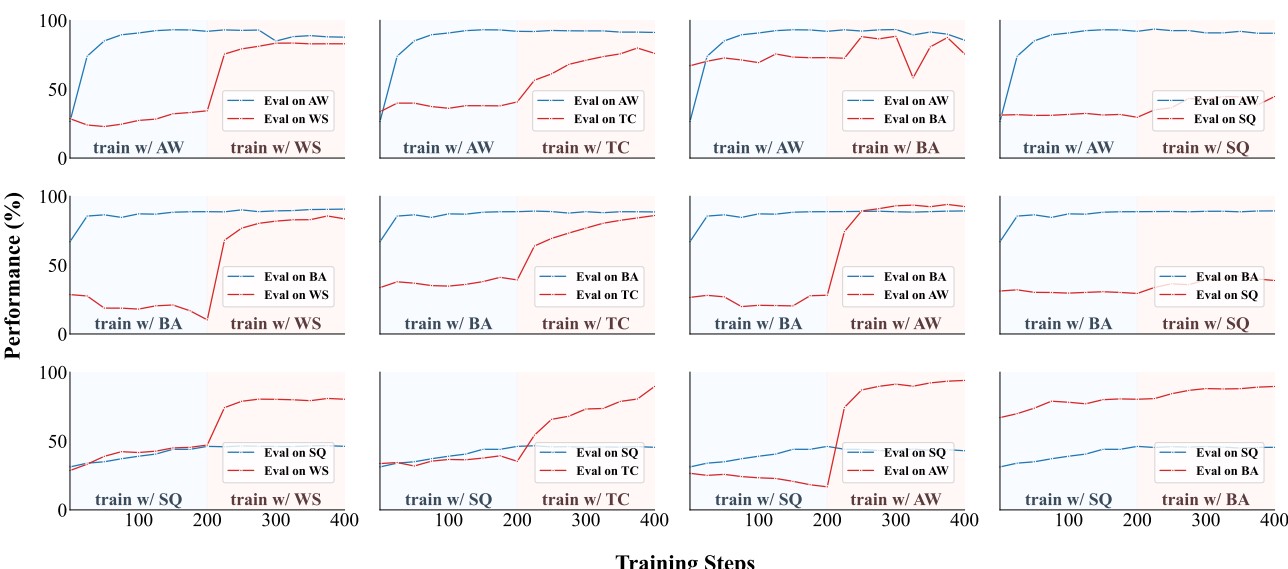

*Figure 7.* Training dynamics of forgetting and transfer in other sequential two-stage cross-environment training with Qwen2.5-7B-Instruct, where blue and red denote the upstream environment and the downstream environment, respectively.

1. **Instruction Misinterpretation**: The agent fails to understand the prompt correctly, does not follow the instructions as indicated, or generates the wrong answer format.

2. **Guessing or Fabrication**: The agent relies too heavily on internal parameters or makes unsupported guesses, instead of utilizing environmental tools to gather necessary external information.

3. **Action Execution Failure**: The agent generates actions that cannot be interpreted by the environment or do not correspond to specific objects within the environment.

4. **Constraint Prioritization Failure**: The agent recognizes multiple constraints in the instructions but fails to evaluate their relative importance correctly, resulting in the violation of core constraints in favor of secondary ones.

5. **Confirmation Bias**: The agent becomes confident that it has found the correct answer or completed key steps, but does not proceed with further verification. Alternatively, it may forcefully apply other clues or validate erroneous information from the prompt.

6. **State or Memory Inconsistency**: The agent exhibits contradictions over time, forgetting tools it has recently invoked or results it has obtained. Completed steps may be repeated unnecessarily.

7. **Logic or Numerical Deficit**: The agent makes errors in logical or numerical reasoning.

8. **Termination Protocol Failure**: The agent fails to issue the environment-specific termination command or prematurely terminates before completing the task.

The results of both in-domain and out-of-domain evaluations are discussed in Section 7. Here, we present a detailed error classification in Figure 9.

## K. Case Study

In this section, we present specific case studies. It is worth noting that while all interactions are inherently text-based, we provide visualizations for selected cases to enhance clarity. Additionally, due to the excessive number of interaction turns, we omit the majority of intermediate steps, highlighting only the pivotal moments in the figures.

Figure 10 illustrates a case where a model trained on SearchQA is evaluated on WebShop, comparing its generated trajectory against that of the base model on the same task. As observed in the case, the base model tends to blindly input the entire

*Table 11.* Results of sequential training across different environments with Qwen2.5-7B-Instruct model.

| UPSTREAM | DOWNSTREAM | WEBSHOP | SEARCHQA | TEXTCRAFT | ALFWORLD | BABYAI |
|---|---|---|---|---|---|---|
| BASE MODEL | | 28.59 | 31.19 | 33.63 | 26.56 | 67.00 |
| WEBSHOP | | 86.50 | 33.28 | 40.75 | 24.13 | 79.21 |
| | + SEARCHQA | 85.08 | 41.44 | 44.63 | 17.81 | 78.88 |
| | + TEXTCRAFT | 86.32 | 36.81 | 82.50 | 26.63 | 81.68 |
| | + ALFWORLD | 85.99 | 34.22 | 43.75 | 90.69 | 76.34 |
| | + BABYAI | 86.38 | 33.97 | 49.25 | 25.50 | 87.74 |
| SEARCHQA | | 47.07 | 46.12 | 35.25 | 16.75 | 80.33 |
| | + WEBSHOP | 80.35 | 46.16 | 34.50 | 18.44 | 78.59 |
| | + TEXTCRAFT | 49.31 | 46.19 | 78.63 | 26.06 | 78.03 |
| | + ALFWORLD | 39.42 | 42.92 | 41.00 | 94.00 | 81.29 |
| | + BABYAI | 44.36 | 45.44 | 25.00 | 14.06 | 89.62 |
| TEXTCRAFT | | 38.30 | 32.19 | 80.88 | 31.50 | 77.95 |
| | + WEBSHOP | 84.33 | 34.59 | 77.00 | 18.31 | 78.44 |
| | + SEARCHQA | 39.17 | 43.16 | 71.63 | 19.69 | 78.97 |
| | + ALFWORLD | 42.04 | 31.56 | 76.50 | 90.38 | 62.19 |
| | + BABYAI | 23.00 | 33.37 | 74.88 | 22.81 | 87.55 |
| ALFWORLD | | 34.31 | 29.59 | 36.13 | 92.00 | 72.91 |
| | + WEBSHOP | 82.95 | 31.50 | 35.88 | 87.75 | 49.65 |
| | + SEARCHQA | 53.04 | 44.66 | 43.50 | 90.63 | 77.86 |
| | + TEXTCRAFT | 50.31 | 30.81 | 76.00 | 91.19 | 77.01 |
| | + BABYAI | 25.68 | 28.31 | 39.25 | 85.56 | 75.59 |
| BABYAI | | 10.25 | 29.41 | 39.25 | 28.13 | 88.79 |
| | + WEBSHOP | 83.60 | 30.19 | 36.38 | 33.38 | 90.66 |
| | + SEARCHQA | 33.71 | 38.84 | 40.63 | 24.06 | 89.37 |
| | + TEXTCRAFT | 17.95 | 33.37 | 86.00 | 29.31 | 88.60 |
| | + ALFWORLD | 13.60 | 29.88 | 41.25 | 92.50 | 89.32 |

content of the instruction into the search bar, resulting in the retrieval of numerous irrelevant items. Furthermore, the base model exhibits incoherent decision-making under multiple constraints and struggles to accurately extract key information from the voluminous HTML content returned by the environment, leading to the selection of items from incorrect categories. In contrast, the model trained on SearchQA learns to formulate more flexible search queries, as well as perform efficient information extraction from complex results, thereby enabling it to successfully retrieve key information and select the correct item.

Figure 11 highlights a key reason why agents struggle to generalize to SearchQA, using model trained on AlfWorld as a case study. Although both models fail in their initial attempts, the model trained on SearchQA demonstrates the ability to refine its search queries to achieve greater precision. This capability proves difficult to transfer from other environments, e.g., the model trained on AlfWorld falls into a repetitive loop of the same searches and answers, consequently failing to resolve the task.

Figure 12 illustrates how training enhances exploration efficiency, using BabyAI as a case study. The base model, lacking strong spatial awareness, struggles to accurately pinpoint the target location based on textual descriptions. Consequently, it falls into a pattern of redundant exploration; although it eventually completes the task, it requires an excessive number of interaction turns. In contrast, the trained model is able to precisely locate the target, thereby completing the task via an optimal path and significantly improving exploration efficiency.

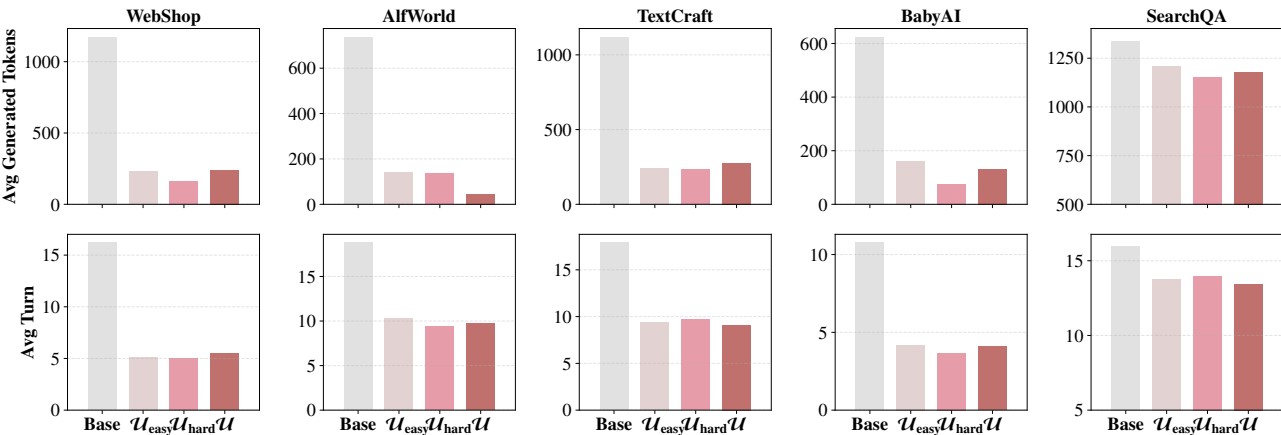

*Figure 8.* Average generated tokens and average turn across different environments for Qwen2.5-3B-Instruct model trained with varying difficulties.

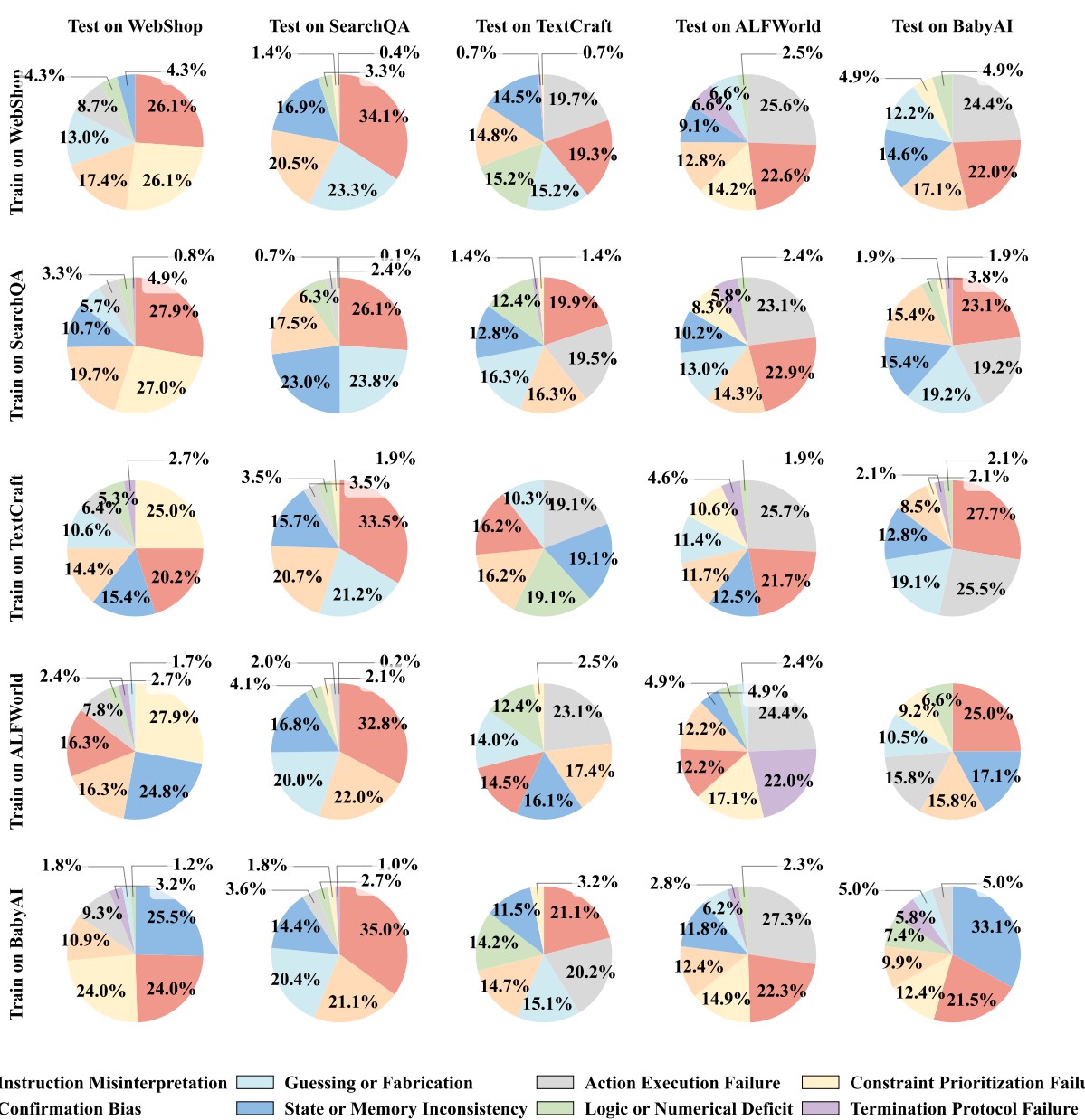

*Figure 9.* Failure mode distribution across all train-test environment combinations.

**WebShop**

**Instruction:** Find me hand wash women's sweaters with long sleeve, stretch fabric, polyester spandex for teen girls, daily wear with color: xnj-tshirt347-white, and size: small, and price lower than 50.00 dollars

**Thought:** I think the best way to find the desired product is to use a search query that includes **all the necessary details** such as gender, style, material, color, and size.
**Action:** search[hand wash women's sweaters long sleeve stretch fabric polyester spandex teen girls daily wear color:xnj-tshirt347-white size:small price:<50.00]

base model

WebShop

WebShop

click[B09QCVCYVY]

base model

< Prev

base model

click[< Prev]
search[…]
…

repeated loop
"search-click-prev"
…

Reaching the maximum number of rounds!

**Score: 0**

**Thought:** I think the search should include the **key details** provided with specific criteria.
**Action:** search[hand wash women's long sleeve sweaters polyester spandex teen girl white small daily wear under 50]

model trained on SearchQA

WebShop

click[B09M63B87V]

model trained on SearchQA

WebShop

color
xnj-tshirt346-wine ⊙ xnj-tshirt347-white ○
size
⊙ small ○ medium ○ large ○ x-large ○ xx-large

click[small]

…

WebShop

Buy Now

model trained on SearchQA

click[Buy Now]

Thank you for shopping with us!

**Score: 1**

*Figure 10.* Comparison of trajectories on WebShop between the base model and the model trained on SearchQA.

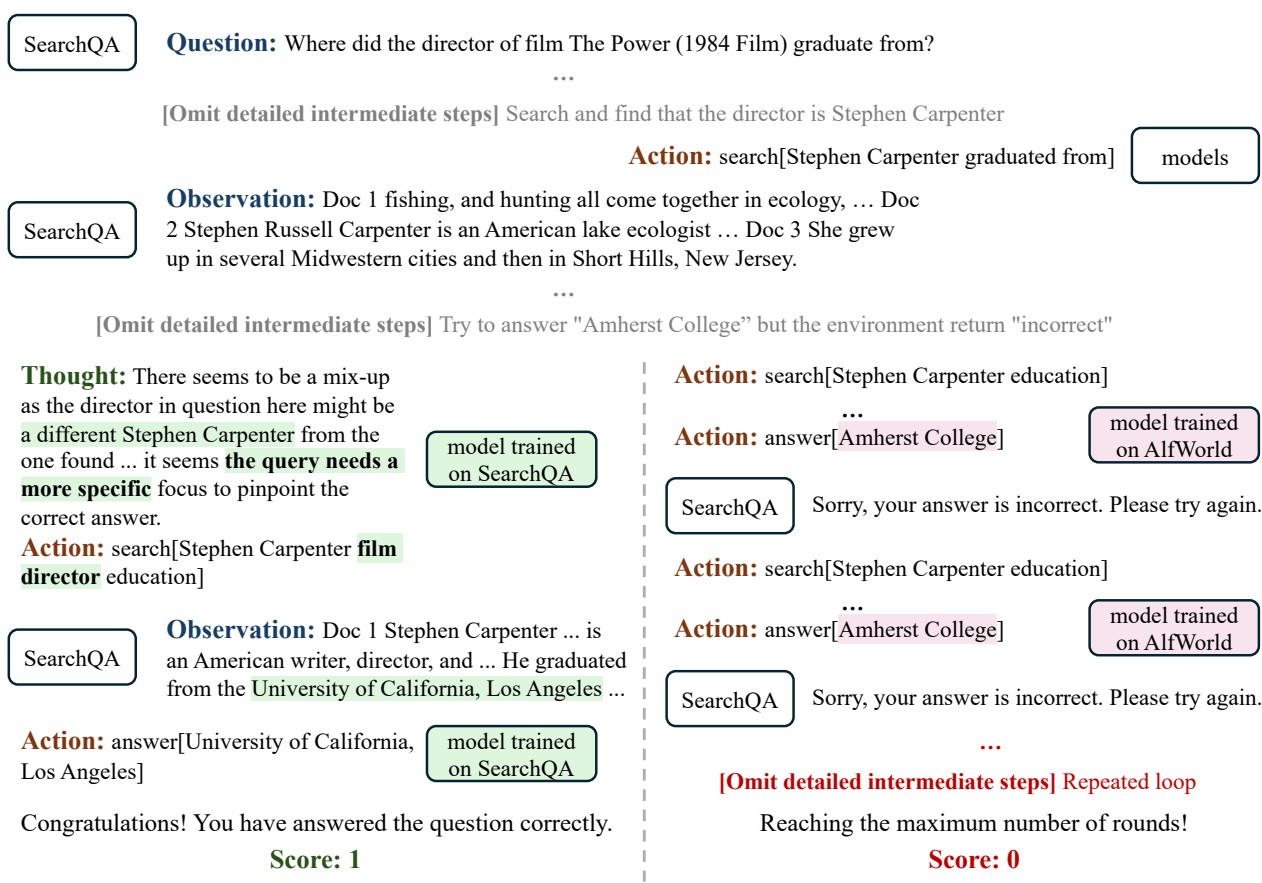

*Figure 11.* Comparison of trajectories on SearchQA between the model trained on SearchQA and the mode trained on AlfWorld.

BabyAI

**Instruction:** go to the yellow ball

In front of you in this room, you can see several objects: There is a red ball 1 0 steps in front of you and 2 steps to your left … The room has walls around you. You are facing a wall 3 steps away. You are not carrying anything.
**Available actions:** ["turn left", "turn right", "move forward", "pickup red ball 1", "pickup green key 1", "pickup red key 1", "pickup green ball 1", "toggle and go through red closed door 1", "go to red ball 1", "go to green key 1", "go to red key 1", "go to yellow locked door 1", "go to green ball 1", "go to red closed door 1", "check available actions"]

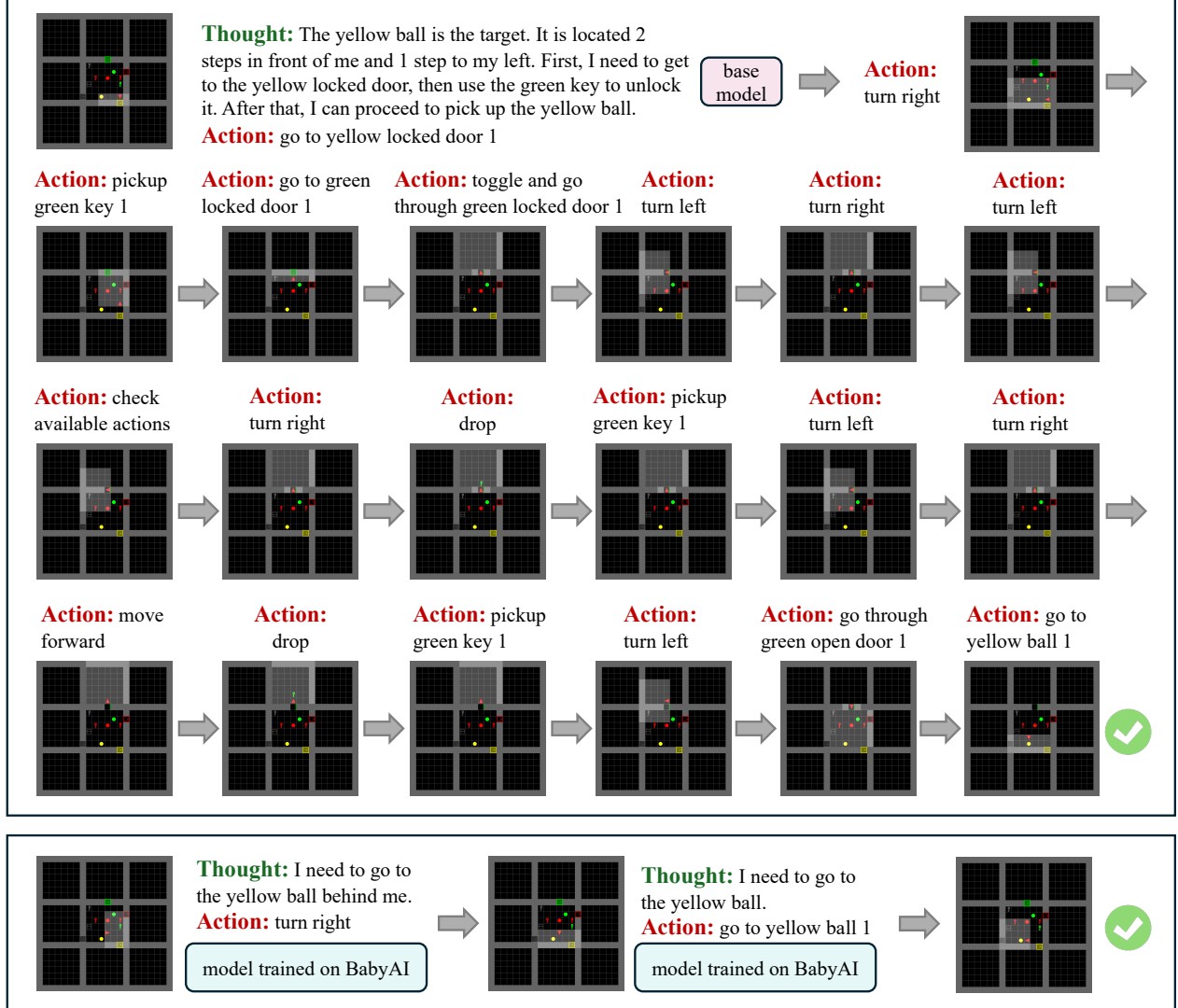

*Figure 12.* A case study in the BabyAI environment. Compared to the base model, the trained model demonstrates significantly improved exploration efficiency.

