# OpenReview forum: "Does Reinforcement Fine-Tuning Improve Generalization of LLM Agents?  An Empirical Study"
_ICML.cc/2026/Conference — ICML 2026 regular_

### Official Review · Reviewer_ezBg · 2026-03-09

**Soundness:** 3
**Presentation:** 3
**Significance:** 3
**Originality:** 3
**Overall Recommendation:** 5
**Confidence:** 4

**Summary:**

The paper studies whether reinforcement fine-tuning (RFT) improves the generalization of LLM agents. The authors conduct a study along 3 axes: (1) within environment generalization across task difficulty, (2) cross-environment transfer to unseen environments, and (3) sequential multi-environment training to quantify transfer and forgetting. The experiments use GRPO to fine-tune Qwen2.5-3B-Instruct and Qwen2.5-7B-Instruct on 5 AgentGym environments. The authors find that RFT transfers well across easy and hard tasks within a fixed environment, yields modest and inconsistent gains in unseen environments, and can support sequential multi-environment training with limited forgetting. The authors then qualitatively analyze and explain the observed behavior.

**Compliance With Llm Reviewing Policy:**

Affirmed.

**Final Justification:**

This is an empirical paper on RFT generalization for LLM agents. Its main strength is the systematic three-axis study design, supported by broad experiments, useful analyses, and clear presentation.

My initial concerns were mostly about robustness of the conclusions and framing relative to prior work. The rebuttal addressed the empirical concerns well with added uncertainty estimates, hyperparameter/stopping ablations, and extra Reinforce++ results. The follow-up also clarified the positioning: the main contribution is best understood as a systematic three-axis characterization of RFT generalization for LLM agents, rather than simply the fact that the setting is multi-turn.

Overall, the rebuttal resolved my main concerns and increased my confidence in the paper. I am therefore raising my score to Accept.

**P.S. Please put the limitations section in the main paper, not the appendix.**

**Key Questions For Authors:**

See Limitations above.

**Limitations:**

Yes, although the limitation section is in the appendix.

**Strengths And Weaknesses:**

Strengths
- This is a very timely study. As LLM relies more on reinforcement post-training nowadays to gain better performance, the amount of generalization and catastrophic forgetting due to RL is important to understand. All 3 questions that the authors pose are very relevant.
- The study is reasonably comprehensive: it spans five environments, two model sizes, three evaluation axes, efficiency metrics, and qualitative examples. The separation between held-in and held-out performance is useful, and the sequential-training analysis is interesting.
- Presentation is also clear: the overall paper is easy to follow, and the tables and figures convey the results well.

Weaknesses
- Given the flood of LLM-related papers, the authors need to better position this paper relative to concurrent or similar studies. Section 2.2 is good, but the authors need to better set up the stage of what has been studied, what is the existing understanding, and what is still missing. The current framing is that prior work studies static and single-turn LLM tasks, and this work studies multi-turn LLM agents and across environments. However, the overall finding seems to be consistent with prior work: RL is relatively good at generalization and avoids forgetting compared to SFT, and that generalization to other tasks really depends on the similarity and coverage. So in what ways is multi-turn different from single-turn, that this study shed lights on? The authors should motivate better. Related, in the discussion section, the authors should anchor the finding in the literature, and discuss what agrees and disagrees with prior findings and what is new. Otherwise, the paper risks being a bit incremental relative to existing literature.
- The study can also go a little deeper and be a bit more comprehensive. For example, related questions that may or may not affect the conclusion include: (1) choice of RL algorithm: the authors only chose GRPO. Would other RL algorithms behave the same? (2) choice of hyperparameters, compute budget, stopping criteria. Would learning rate affect things? Would over-training lead to different conclusions? (3) Would LoRA have different behaviors compared to full finetuning? Going deeper in some of these axes could make the paper stronger.
- The study should report uncertainty estimates/confidence interval on the numbers. This would help qualify the conclusion better.
- Some relevant papers to consider citing:

[1] Gu, L., Jiang, Z., Chi, Z., Liu, H., Wang, Z., Yu, Y., Berseth, G., and Wang, Y. Generalization in Online Reinforcement Learning for Mobile Agents. 2025.

[2] Szot, A., Schwarzer, M., Agrawal, H., Mazoure, B., Metcalf, R., Talbott, W., Mackraz, N., Hjelm, R. D., and Toshev, A. T. Large Language Models as Generalizable Policies for Embodied Tasks. ICLR 2023.

[3] Zhang, H., Liu, X., Lv, B., Sun, X., Jing, B., Iong, I. L., Qi, Z., Lai, H., Xu, Y., Lu, R., Hou, Z., Wang, H., Tang, J., and Dong, Y. AgentRL: Scaling Agentic Reinforcement Learning with a Multi-Turn, Multi-Task Framework. arXiv 2025.

[4] Xia, Y., Fan, J., Chen, W., Yan, S., Cong, X., Zhang, Z., Lu, Y., Lin, Y., Liu, Z., and Sun, M. AgentRM: Enhancing Agent Generalization with Reward Modeling. In Proceedings of the 63rd Annual Meeting of the Association for Computational Linguistics (Volume 1: Long Papers), 2025.

[5] Chen, H., Razin, N., Narasimhan, K. R., and Chen, D. Retaining by Doing: The Role of On-Policy Data in Mitigating Forgetting. arXiv 2025.

[6] Liu, Z., Guan, L., Nie, Y., Zhang, K., Hao, Z., Chen, L., ... & Zhang, N. Paying Less Generalization Tax: A Cross-Domain Generalization Study of RL Training for LLM Agents. arXiv 2026.

---

> ### Author Rebuttal · Authors · 2026-03-31
>
> We appreciate your valuable feedback, and for highlighting our strengths (e.g., **high practical significance, comprehensive experiments, and clear presentation**). In the following, we will carefully respond to your questions.
>
> ### **# Question about different hyperparameters and stopping criteria**
>
> Following your suggestion, we conducted ablation experiments on **Qwen2.5-7B-Instruct** to study the effects of learning rate and rollout_n (set to **1e-6** and **8** in the paper), and also evaluated stopping at different training steps (set to **200** in the paper). The results in the tables below show that different hyperparameter settings have only a small effect. Although more sufficient training and more careful hyperparameter tuning may lead to better performance, they do not affect our overall conclusions.
>
> |Train w/|lr|rollout_n|WebShop|SearchQA|TextCraft|AlfWorld|BabyAI|Avg|
> |-|-|-|-|-|-|-|-|-|
> |WebShop|1e-6|8|86.50|33.28|40.75|24.13|79.21|52.8|
> ||1e-6|4|84.03|31.87|37.50|24.56|78.69|51.3|
> ||2e-6|8|86.09|30.50|36.00|25.41|78.17|51.2|
> ||5e-7|8|82.86|33.66|33.50|25.94|76.28|50.5|
> |SearchQA|1e-6|8|47.07|46.12|35.25|16.75|80.33|45.1|
> ||1e-6|4|48.53|42.19|33.50|17.12|78.25|43.9|
> ||2e-6|8|44.31|40.10|29.88|20.38|77.70|42.5|
> ||5e-7|8|42.51|40.22|35.38|23.25|75.70|43.4|
>
>
>
> |Train w/|Step|WebShop|SearchQA|TextCraft|AlfWorld|BabyAI|Avg|
> |-|-|-|-|-|-|-|-|
> |WebShop|100|86.13|34.00|36.13|23.63|77.24|51.4|
> ||150|85.59|34.03|38.00|23.75|78.18|51.9|
> ||200|86.50|33.28|40.75|24.13|79.21|52.8|
> |SearchQA|100|41.65|38.97|36.62|23.31|78.15|43.7|
> ||150|44.99|44.00|37.63|20.75|80.02|45.5|
> ||200|47.07|46.12|35.25|16.75|80.33|45.1|
>
> ### **# Question about confidence interval of results**
>
> Thank you for your suggestions. We report the confidence interval in the table below (Qwen2.5-3B-Instruct), demonstrating the stability of our conclusions.  Others will be added into the next version of the manuscript.
>
> |Train w/|WebShop|SearchQA|TextCraft|AlfWorld|BabyAI|
> |-|-|-|-|-|-|
> |WebShop|87.86±0.18|25.84±0.47|18.50±1.50|23.06±0.96|70.91±0.48|
> |SearchQA|22.97±1.23|41.78±0.20|22.75±1.50|12.13±0.25|65.72±2.81|
> |TextCraft|14.46±0.26|22.16±0.38|73.88±1.00|11.00±0.63|63.73±1.04|
> |AlfWorld|21.86±0.35|23.50±1.05|24.88±2.06|91.81±0.41|64.68±1.59|
> |BabyAI|28.36±0.65|22.63±0.83|16.75±0.96|4.50±0.41|86.55±1.03|
>
> ### **# Question about more RL algorithms**
>
> Following your suggestions, we conducted experiments of **Reinforce++** algorithm in both within-environment and cross-environment scenarios. Due to word limits, we are unable to present detailed results here. Please refer to our response to **Reviewer XnU7** for more information. We sincerely apologize for any inconvenience caused.
>
> ### **# Question about comparison with similar studies**
>
> Thank you very much for your suggestion. We will discuss these issues in greater detail in the revised manuscript.
>
> In brief, our work provides a **systematic study** of agent generalization from three perspectives, which distinguishes it from prior work. Although previous studies such as Mobile Agent [1] and Embodied Agent [2] have also examined generalization in reinforcement learning settings, they remain largely confined to a single task type.
>
> For **within-environment generalization**, we focus on transfer across tasks with different difficulty levels and demonstrate the effectiveness of curriculum learning from a generalization perspective. For **cross-environment generalization**, we provide a more in-depth analysis of transfer across different source and target environments. AgentRL [3] investigates mixed-training experiments, and it does not focus on generalization itself or provide a detailed analysis of the underlying dynamics. For **sequential training**, we go beyond studying forgetting alone. While Retaining by Doing [4] mainly focuses on forgetting, we additionally examine transferability and the cross-environment generalization behavior that emerges under sequential training, and further establish a connection between cross-environment generalization and sequential-training generalization. In addition, we conduct a detailed failure analysis to highlight the key challenges that remain for robust agent generalization.
>
> ---
>
> We once again thank you for your valuable suggestions. If you have any further questions, please feel free to let us know and we will do our utmost to respond. If you find our replies satisfactory, we kindly ask that you consider updating your score and confidence accordingly.
>
> **References**
>
> [1] Generalization in Online Reinforcement Learning for Mobile Agents
>
> [2] Large Language Models as Generalizable Policies for Embodied Tasks
>
> [3] AgentRL: Scaling Agentic Reinforcement Learning with a Multi-Turn, Multi-Task Framework
>
> [4] Retaining by Doing: The Role of On-Policy Data in Mitigating Forgetting

---

> > ### Author Rebuttal · Reviewer_ezBg · 2026-04-04
> >
> > Thank you for the detailed rebuttal. The added ablations, uncertainty estimates, and Reinforce++ results resolve most of my concrete concerns.
> >
> > My remaining concern is mainly about framing. The readers should be able to crisply understand and agree what is lacking before your study and what is new. The current paper leans on the multi-turn setting as the main differentiating angle, but I am not yet fully convinced that this distinction is made sharply enough relative to prior work. In the rebuttal, the positioning seems to partially pivot toward the paper being a systematic three-axis empirical study of agent generalization. That may be fine, but if so, the paper should make that choice more explicit and more carefully distinguish what is already consistent with prior literature from what is newly established here.
> >
> > I would therefore encourage the authors to clarify what they see as the primary contribution, the multi-turn angle, the three-axis study, or something else, and to sharpen the positioning accordingly. If this is clarified, I would be comfortable bumping up from Weak Accept to Accept.

---

> > > ### Author Response · Authors · 2026-04-06
> > >
> > > Thank you very much for your helpful suggestions and encouragement. We deeply appreciate your constructive feedback, which has been instrumental in improving the framing of our paper. We completely agree that the positioning should be more explicit and sharp.
> > >
> > > In the revision, we will clarify that our primary contribution is not simply that the setting is multi-turn. **Rather, our main contribution is a systematic three-axis empirical characterization of how RFT generalizes for LLM agents, with multi-turn interaction being the crucial setting where this question becomes practically meaningful.** Specifically, for interactive agents, generalization is not only about solving new instances from a similar distribution (as in static tasks); it fundamentally requires transferring policies under shifts in world knowledge, observation spaces, action interfaces, and error-recovery dynamics. These challenges are much weaker or entirely absent in static or single-turn settings.
> > >
> > > Additionally, we will more explicitly separate what is consistent with prior literature from what is newly established in our study:
> > >
> > > 1. Consistent with prior work: Our findings align with prior post-training research showing that RL generally preserves capabilities better than SFT, and that successful transfer heavily depends on task similarity and behavioral coverage.
> > >
> > > 2. Newly established by our work: The novel contribution of our paper is demonstrating how this generalization profile decomposes differently across our three proposed agent-specific axes. We uniquely reveal: (1) strong within-environment difficulty transfer driven by curriculum learning effects; (2) substantially weaker transfer to unseen environments due to sensitivity to knowledge and interface shifts; and (3) promising sequential multi-environment training dynamics that yield downstream transfer with minimal catastrophic forgetting. Crucially, we go beyond merely reporting performance metrics by conducting in-depth qualitative analyses of the underlying error modes, thereby providing actionable insights for developing more robust agents.
> > >
> > > **To further sharpen our positioning relative to the prior work you highlighted [1-6] in the original review, we will explicitly discuss how our paper is complementary but distinct:**
> > >
> > > - Generalization studies in mobile or embodied RL settings [1, 2] analyze transfer within a single, specific embodied domain. In contrast, our work compares five highly heterogeneous agent environments and studies three distinct deployment-relevant axes within a unified framework.
> > >
> > > - AgentRL [3] primarily introduces a framework for scaling multi-turn, multi-task agent RL. Our goal is not to introduce a new training recipe, but to deeply measure when and why standard RFT generalizes or fails, specifically analyzing source-target asymmetry and the impact of knowledge/interface shifts.
> > >
> > > - AgentRM [4] proposes reward modeling to improve agent generalization. In contrast, we characterize the baseline generalization profile of standard RFT itself, thereby identifying the exact bottlenecks where such additional methods are most needed.
> > >
> > > - Retaining by Doing [5] focuses on mitigating forgetting through on-policy data. We go beyond forgetting to jointly analyze transfer and forgetting under sequential multi-environment training, explicitly connecting these dynamics to cross-environment generalization.
> > >
> > > - The concurrent work on cross-domain generalization tax [6] **(released on arXiv on January 26, just two days before the ICML full paper submission deadline on January 28)** is perhaps the closest in spirit, and we will discuss it explicitly. While their study focuses on cross-environment generalization across 4 environments (compared to our 5), our work differs by providing a unified three-axis study. Beyond evaluating cross-environment transfer, we additionally cover within-environment difficulty transfer and sequential/mixed multi-environment training, together with qualitative analyses tying failures to shifts in knowledge and interfaces.
> > >
> > > Following your guidance, we will revise the manuscript to make this framing much sharper. Due to the 5,000-character limit of this response, we will provide a more detailed and comprehensive discussion in the revision. We will explicitly position the paper as a systematic empirical characterization of agent generalization, clarify the specific role of the multi-turn angle, and clearly demarcate our novel findings from existing literature. Thank you again for helping us refine the core message of our work. We hope these clarifications address your remaining concerns and that you will consider raising your score to an Accept!

---

### Official Review · Reviewer_CHDi · 2026-03-10

**Soundness:** 3
**Presentation:** 3
**Significance:** 2
**Originality:** 2
**Overall Recommendation:** 3
**Confidence:** 4

**Summary:**

This paper employs the AgentGym environment to train agents via AgentGym-RL, and mainly investigates the performance of agents in cross-difficulty and cross-task generalization.

**Compliance With Llm Reviewing Policy:**

Affirmed.

**Key Questions For Authors:**

1. Specific considerations for only selecting AgentGym and AgentGym-RL
2. What work can be done based on the research in this paper to explore the generalization ability of agents in real-world environments

**Limitations:**

yes

**Strengths And Weaknesses:**

The empirical work is comprehensive and detailed, including failure mode analysis and case studies, and the paper is highly readable.

My main concern is that the innovation of this work is not sufficiently strong, for the following specific reasons:
1. Although the performance and analysis of the agent across task difficulties and tasks align with mainstream viewpoints, no novel perspectives are presented.
2. The empirical analysis in the paper is mainly conducted based on two related works—the AgentGym benchmark and the AgentGym-RL training framework—without exploring other benchmarks or training frameworks, which may cast doubt on the generality of the conclusions.

In addition, I have some concerns about several overly absolute claims in the paper, such as:
1. Most benchmarks are mainly limited to within-domain evaluation; is there support from relevant surveys? In other words, is work on generalization evaluation scarce?
2. Most existing studies focus on static, single-turn large language model tasks; is there support from relevant surveys?
3. The underlying assumption is that the conclusions obtained from the controlled AgentGym environment in this paper can provide references for real-world deployment. However, conclusions from a single simulated environment like AgentGym may not be fully transferable.
4. Due to the large workload involving multiple experiments, the paper does not conduct multiple independent evaluations or report the stability of the conclusions.

---

> ### Author Rebuttal · Authors · 2026-03-31
>
> We appreciate your valuable time, feedback, and for highlighting our strengths (e.g., **comprehensive and detailed experienments, and highly readable writing**). In the following, we will carefully respond to your questions.
>
> ### **# Question about the stability of conclusions**
>
> Thank you for your valuable feedback! First, we would like to clarify that, we performed 8 samplings for each evaluation to reduce randomness, which is more robust than avg@4 of other studies [1]. Following your suggestions, we also report the evaluation error in the table below, which demonstrates the stability. We will add the results to the next version of the manuscript!
>
> |Train w/|WebShop|SearchQA|TextCraft|AlfWorld|BabyAI|
> |-|-|-|-|-|-|
> |WebShop|87.86±0.18|25.84±0.47|18.50±1.50|23.06±0.96|70.91±0.48|
> |SearchQA|22.97±1.23|41.78±0.20|22.75±1.50|12.13±0.25|65.72±2.81|
> |TextCraft|14.46±0.26|22.16±0.38|73.88±1.00|11.00±0.63|63.73±1.04|
> |AlfWorld|21.86±0.35|23.50±1.05|24.88±2.06|91.81±0.41|64.68±1.59|
> |BabyAI|28.36±0.65|22.63±0.83|16.75±0.96|4.50±0.41|86.55±1.03|
>
> ### **# Question about framework selection**
>
> Thank you very much for your kind suggestion. We would like to clarify that AgentGym is not a single simulation environment, but rather a framework that integrates multiple environments and tasks, which aligns well with our goal of studying within-environment, cross-environment, and sequential-training settings. In fact, most existing studies are conducted within a single training framework, largely due to computational resource constraints. In future work, we hope to follow your suggestion and validate our findings across additional frameworks, such as AgentRL [2].
>
> ### **# Question about novel perspectives and supporting claims**
>
> Thank you for your suggestion. We would first like to clarify that the current evaluation of LLM agent generalization remains limited, and our work is intended to help fill this gap. Although there has been prior work on the generalization of LLM reasoning, such studies do not consider the dynamic, multi-turn interactions between agents and environments. Owing to space constraints, we were unable to discuss many related works here, but we will incorporate a more comprehensive review of the literature and better position our contributions in the revised manuscript.
>
> In brief, we aims at conducting a **systematic study** of agent generalization from three perspectives. First, for **within-environment generalization**, we investigate transfer across tasks with different difficulty levels and show the effectiveness of curriculum learning from a generalization standpoint. Second, for **cross-environment generalization**, we provide an in-depth analysis of how generalization emerges across different source and target environments. This differs from prior work such as Mobile Agent [3] and Embodied Agent [4], which mainly focus on generalization within a single task type. Third, for **sequential training**, we go beyond studying forgetting alone, as explored in Retaining by Doing [5], by also analyzing transfer effects and broader generalization behavior in sequential training, and further connecting these findings to cross-environment generalization.
>
> Beyond these three settings, we also conduct a detailed failure analysis to better reveal the key challenges of agent generalization.
>
> ### **# Question about what work can be done based on our research**
>
> Thank you for your forward-looking question. Recent systems such as Claude Code and OpenClaw demonstrate how agents interact with complex, evolving, real-world environments. Here, we identify several promising research directions:
>
> (1) Developping methods to improve agent generalization
>
> Based on our analysis, generalization remains a serious challenge. To deploy agents in the real world, more effective solutions are needed, i.e., finding appropriate scaling dimensions and algorithms.
>
> (2) Lifelong learning
>
> Future work in real world can extend our sequential multi-environment training to continual learning, where agents must balance transfer and forgetting over long time horizons.
>
> (3) Scalable oversight
>
> As agents continue to grow in capability and autonomy in real world, how to supervise them and ensure that they develop in a safe and controllable way has become important.
>
> ---
>
> We once again thank you for your valuable comments! If you have any further questions, please feel free to let us know and we will do our utmost to respond. If you find our replies satisfactory, we kindly ask that you consider updating your score accordingly.
>
> **References**
>
> [1] Beyond Ten Turns: Unlocking Long-Horizon Agentic Search with Large-Scale Asynchronous RL
>
> [2] AgentRL: Scaling Agentic Reinforcement Learning with a Multi-Turn, Multi-Task Framework
>
> [3] Generalization in Online Reinforcement Learning for Mobile Agents
>
> [4] Large Language Models as Generalizable Policies for Embodied Tasks
>
> [5] Retaining by Doing: The Role of On-Policy Data in Mitigating Forgetting

---

> > ### Author Rebuttal · Reviewer_CHDi · 2026-04-04
> >
> > I recognize that the author has supplemented experiments on stability analysis. However, I believe the issue of research innovation remains a key drawback of this paper. This is because comparable results to those presented in this paper can also be achieved by adopting the AgentGym evaluation framework and the AgentGym-RL training framework. In my opinion, the current innovation of this work merely achieves the effect of one plus one equals two, and I would prefer to see research that delivers outcomes where one plus one is greater than two.
> >
> > The core of this concern lies in that the exploration of generalization should not be limited to a single previous study. AgentGym is an existing research achievement, and AgentGym-RL is a framework developed for training models within this environment. If readers aim to obtain novel conclusions on generalization, relying solely on this paper will only allow them to draw analyses specific to AgentGym. When these analyses are applied to readers' customized evaluation and training methods, they may even lead to contradictory conclusions. Although AgentGym is a hybrid environment, readers cannot identify whether the conclusions derived from this environment are biased, which may mislead them.
> >
> > Nevertheless, I appreciate the author's writing style, so I will retain the original score.

---

> > > ### Author Response · Authors · 2026-04-06
> > >
> > > Thank you very much for your follow-up comments. We appreciate your recognition, and we would like to further clarify our position.
> > >
> > > We believe the novelty of this work lies not in proposing a new benchmark, algorithm, or training framework, but in the research motivation and the empirical study design used to address it. Our central motivation is to understand how reinforcement fine-tuning affects the transferability and generalization of LLM agents. To study this question, we design a unified multi-environment empirical framework and systematically investigate three complementary settings: (i) within-environment transfer across task difficulty, (ii) cross-environment transfer to unseen environments, and (iii) sequential multi-environment training with transfer and forgetting analysis. In this sense, the contribution of our paper is a new empirical study design for characterizing RFT generalization of LLM agents, together with the resulting insights. **These questions, and the insights derived from them, are not automatically obtained from "AgentGym plus AgentGym-RL".**
> > >
> > > We chose AgentGym and AgentGym-RL because they provide a mature and suitable infrastructure with diverse environments and a unified training protocol. **In our view, studying a new research question on top of an existing benchmark or framework should not necessarily be seen as lacking novelty.** Indeed, many prior works have investigated the properties of RL using existing frameworks such as VERL, Slime, and OpenRLHF. We believe such studies, like ours, can still make meaningful research contributions.
> > >
> > > We also understand that your concern may partly relate to external validity. We fully agree that validating the findings on additional frameworks would be valuable and would further strengthen the paper. At the same time, we believe that research conducted on a mature and well-established framework is also worthwhile. **In fact, many current LLM Agent research papers are built on a single Agent framework with few environments/tasks.** For example, *"The Lighthouse of Language: Enhancing LLM Agents via Critique-Guided Improvement"* was accepted as a NeurIPS 2025 poster and is based on the AgentGym setup, while including only three environments. In addition, recent work such as *"AgentPRM"* also continues this line of research on top of AgentGym.
> > >
> > > **Again, we sincerely appreciate your suggestions and feedback, and we will do our best to improve the manuscript.**

---

### Official Review · Reviewer_XnU7 · 2026-03-11

**Soundness:** 4
**Presentation:** 3
**Significance:** 4
**Originality:** 3
**Overall Recommendation:** 5
**Confidence:** 4

**Summary:**

This paper presents a comprehensive empirical study on the generalization capabilities of Reinforcement Fine-Tuning (RFT) for Large Language Model (LLM) agents. Moving beyond standard in-domain evaluations, the authors systematically investigate RFT performance along three critical axes: (1) intra-environment generalization across varying task difficulties, (2) inter-environment transfer to unseen environments with different observation/action spaces, and (3) sequential multi-environment training to analyze transfer and forgetting dynamics. Experiments conducted across five diverse benchmarks (WebShop, SearchQA, TextCraft, AlfWorld, BabyAI) reveal that while RFT generalizes well within environments and benefits from curriculum learning, its transfer to unseen environments is sensitive to shifts in semantic priors and interface structures. Furthermore, the study demonstrates that sequential training can achieve downstream gains with minimal upstream forgetting, often matching the performance of joint mixture training. The paper provides valuable insights into failure modes and offers practical guidance for deploying generalizable LLM agents.

**Compliance With Llm Reviewing Policy:**

Affirmed.

**Final Justification:**

This paper makes a strong empirical contribution by systematically studying how reinforcement fine-tuning affects the generalization of LLM agents along three complementary axes: within-environment difficulty transfer, cross-environment transfer, and sequential multi-environment training. The experimental scope is broad, covering five environments with different properties, and the paper goes beyond aggregate scores by providing training-dynamics analysis, failure-mode breakdowns, and case studies that help explain when transfer succeeds or fails. In particular, the results show strong within-environment transfer and useful curriculum effects, while also revealing that out-of-environment generalization is much more sensitive to shifts in semantic priors and interfaces.

My original concerns were relatively minor and mainly about presentation polish and the generality of the conclusions beyond GRPO. The rebuttal addressed these points well by clarifying that the main tables report success rates, committing to fix the presentation issues, and providing additional Reinforce++ experiments that show broadly similar trends, which strengthens confidence that the main observations are not purely specific to GRPO. The paper’s core conclusions therefore remain well supported, and I keep my positive assessment and original score.

**Key Questions For Authors:**

1.  **Typo in Abstract**: In the abstract, the phrase "In real-worlddeployment" is missing a space. It should be "In real-world deployment". Please proofread the entire manuscript for similar minor spacing errors.
2.  **Clarification of Metrics in Tables**: In Table 2 and Table 3, the numerical values represent performance, but the headers or captions do not explicitly state whether these are **Success Rates (%)** or raw **Scores**. Given that some environments use binary rewards and others use dense scores (as mentioned in Section 3.2), please clearly label the units in the table captions or column headers to avoid ambiguity.
3.  **Definition of Abbreviations in Figures**: In Figure 3, the labels "TC", "WS", "SQ", "AW", and "BA" are used without definition in the caption or the immediate vicinity of the figure. While they are defined later in Figure 4's caption, readers encountering Figure 3 first may be confused. Please define these abbreviations in the caption of Figure 3 to ensure self-containment.

**Limitations:**

Yes.

**Strengths And Weaknesses:**

### Strengths
1.  **Systematic and Comprehensive Evaluation**: The three-axis framework (difficulty, cross-environment, sequential training) provides a holistic view of RFT generalization that is largely missing in current literature. The scale of the evaluation (5 environments, 2 model sizes, extensive sequential combinations) is impressive and yields robust conclusions.
2.  **High Practical Significance**: The findings directly address a critical gap between research settings (in-domain) and real-world deployment (distribution shift). Insights such as the sensitivity of transfer to action space changes (e.g., BabyAI's available action list causing dependency) and the effectiveness of easy-to-hard curriculum learning are highly actionable for practitioners.
3.  **Deep Analysis and Insights**: Beyond reporting metrics, the paper offers deep qualitative analysis, including detailed failure mode categorization (Figure 6) and illuminating case studies (Appendix G) that explain *why* transfer succeeds or fails. The comparison between sequential and joint training is particularly valuable for understanding continual learning dynamics in agents.
4.  **Clarity of Presentation**: The paper is generally well-written, with a logical flow that guides the reader through complex experimental setups. The visualizations (e.g., heatmaps in Figure 4, training dynamics in Figure 3/5) effectively communicate the key trends.

### Weaknesses
1.  **Minor Presentation Oversights**: Despite the high quality of content, there are several avoidable presentation errors that detract from the polish of the submission. These include typos in the abstract, ambiguous metric definitions in tables, and insufficient explanations of abbreviations in figure captions.
2.  **Limited Algorithmic Diversity**: The study primarily focuses on GRPO. While justified as a popular choice, briefly discussing whether the observed generalization patterns might differ with other RL algorithms (e.g., PPO, REINFORCE++) or providing a sentence on this limitation would strengthen the generality of the claims.

---

> ### Author Rebuttal · Authors · 2026-03-31
>
> We appreciate your valuable time, feedback, and positive assessment of our work, particularly **systematic and comprehensive evaluation, high practical significance, deep analysis and insights, and clarity of presentation**. In the following, we will carefully respond to your questions.
>
> ### **# Question about more algorithms**
>
> Thank you for your kind reminder. Following your suggestion, we conducted additional experiments using Reinforce++. First, we performed cross-environment experiments and found that Reinforce++ improves performance in the held-in environment. In terms of cross-environment generalization, Reinforce++ still demonstrates a certain degree of transfer capability, but its performance remains inferior to that of GRPO. These findings are consistent with prior studies [1-3], and they also constitute one of the main reasons why we chose GRPO.
>
> |Train w/|Method|TextCraft|AlfWorld|BabyAI|
> |-|-|-|-|-|
> |Base model||33.63|26.56|67.00|
> |TextCraft|Reinforce++|48.25|22.63|71.04|
> ||GRPO|80.88|31.50|77.95|
> |AlfWorld|Reinforce++|34.88|45.44|69.86|
> ||GRPO|36.13|92.00|72.91|
> |BabyAI|Reinforce++|36.25|21.44|73.68|
> ||GRPO|39.25|28.13|88.79|
>
> In addition, we validated the Reinforce++ algorithm in the BabyAI environment. The results in the table below show that Reinforce++ exhibits patterns similar to those of GRPO: it demonstrates strong transferability across different difficulty levels, and curriculum learning can further improve its performance.
>
> |Train w/|easy|hard|all|
> |-|-|-|-|
> |Base model|80.96|47.90|67.00|
> |U_easy|89.04|60.17|76.85|
> |U_hard|91.44|63.03|79.44|
> |U_easy+U_hard|93.84|69.44|83.54|
>
> Due to time and resource constraints, we leave more detailed experiments to the next version of the paper. We sincerely thank you again for your constructive feedback.
>
> ### **# Question about presentation oversights**
>
> Thank you for your detailed feedback. We sincerely apologize for the presentation issues you pointed out, and will recheck the entire manuscript to correct them in the next version. Regarding the numerical values, we consistently report success rates in Table 2 and Table 3. We apologize for the ambiguity and will clarify it in our manuscript.
>
> ---
>
> We once again thank you for your valuable comments! Your suggestions have provided us with a new perspective for explanation and analysis. If you have any further questions, please feel free to let us know and we will do our utmost to respond.
>
> **References**
>
> [1] AgentGym-RL: Training LLM Agents for Long-Horizon Decision Making through Multi-Turn Reinforcement Learning
>
> [2] On the Interplay of Pre-Training, Mid-Training, and RL on Reasoning Language Models
>
> [3] Breaking Barriers: Do Reinforcement Post Training Gains Transfer To Unseen Domains?

---

> > ### Author Rebuttal · Reviewer_XnU7 · 2026-04-02
> >
> > Thank you for the rebuttal and for the additional Reinforce++ results. I will keep my current score.

---

> > > ### Author Response · Authors · 2026-04-06
> > >
> > > We are delighted to have resolved your concerns. Your suggestions help us further enhance the quality of the manuscript. Thank you once again for your valuable time and recognition. And we will carefully revise the manuscript following your feedback.

---

### Official Review · Reviewer_cMcU · 2026-03-12

**Soundness:** 3
**Presentation:** 3
**Significance:** 3
**Originality:** 3
**Overall Recommendation:** 4
**Confidence:** 4

**Summary:**

The paper evaluates whether reinforcement fine-tuning (RFT) actually improves the generalization of LLM agents, rather than just their in-domain performance. A broad area examined by this study is transfer and forgetting in multi-turn agent RL across different environments, task difficulties, and sequential multi-environment training. The paper studies five environments, WebShop, SearchQA, TextCraft, AlfWorld, and BabyAI, and organizes the experiments along three axes: within-environment generalization, cross-environment transfer, and sequential multi-environment training.

**Compliance With Llm Reviewing Policy:**

Affirmed.

**Final Justification:**

I maintain my score.

**Key Questions For Authors:**

1. Can you disentangle which factor matters most for cross-environment transfer: semantic prior shift, observation format shift, or action-space shift?
2. How sensitive are the conclusions to the choice of avg@8? Do the same trends hold under single-sample evaluation?

**Limitations:**

No, see above

**Strengths And Weaknesses:**

Strength:
1. The three-axis framework is clean and easy to follow: intra-environment difficulty transfer, inter-environment transfer, and sequential multi-environment training.
2. The empirical trends are interesting. In Table 2, RFT clearly improves performance across difficulty levels within the same environment, and easy-to-hard curriculum often helps further.
3. The paper does not claim universal transfer; instead, it shows modest held-out gains on average and even negative transfer in some cases, especially involving BabyAI. That makes the study feel more honest.

Concerns:
1. The paper is still purely empirical, and the explanatory claims are a bit stronger than the evidence. For example, several conclusions are attributed to differences in prior knowledge, action validation strictness, or feedback density, but these are mostly post-hoc interpretations rather than being isolated by controlled ablations.
2. The evaluation metric choice is a bit awkward. The paper reports avg@8 as the main metric, which is reasonable for reducing randomness, but it also makes the reported gains less directly interpretable for practical single-rollout deployment. PASS@1-like metrics would be useful as the main view too.
3. The failure analysis relies on GPT-5-mini to categorize trajectories. That is a practical choice, but it would be better to include annotation validation or inter-rater agreement, since some error categories may be subjective.
4. The comparison is missing a stronger baseline dimension. Since the paper is about whether RFT improves generalization, it would be helpful to compare against supervised fine-tuning or other post-training strategies under matched data settings, not just against the base model.

---

> ### Author Rebuttal · Authors · 2026-03-31
>
> We appreciate your valuable time, feedback, and for recognizing our strengths (e.g., **the clarity and comprehensiveness of our three-axis framework, the interesting and nuanced insights from our empirical findings**). In the following, we will carefully respond to your questions.
>
> ### **# Question about evaluation metric avg@8**
>
> First, we would like to clarify that our 8 evaluations are independent, with each result representing PASS@1 performance. To reduce randomness, we chose avg@8 as the main metric. Regarding your concern about single-sample evaluation, we report the standard deviation across multiple samples in the table below (Qwen2.5-3B-Instruct). The results demonstrate high stability of our evaluation numbers. Other results will be added into the next version of our manuscript.
>
> |Train w/|WebShop|SearchQA|TextCraft|AlfWorld|BabyAI|
> |-|-|-|-|-|-|
> |WebShop|87.86±0.18|25.84±0.47|18.50±1.50|23.06±0.96|70.91±0.48|
> |SearchQA|22.97±1.23|41.78±0.20|22.75±1.50|12.13±0.25|65.72±2.81|
> |TextCraft|14.46±0.26|22.16±0.38|73.88±1.00|11.00±0.63|63.73±1.04|
> |AlfWorld|21.86±0.35|23.50±1.05|24.88±2.06|91.81±0.41|64.68±1.59|
> |BabyAI|28.36±0.65|22.63±0.83|16.75±0.96|4.50±0.41|86.55±1.03|
>
> ### **# Question about more baselines**
>
> Following your suggestions, we conducted experiments on Qwen2.5-7B-Instruct using SFT. We found that **GRPO outperforms SFT on held-in environments** and **exhibits better generalization performance on other environments**, further validating the advantages of RFT.
>
> |Train w/|Method|WebShop|SearchQA|TextCraft|AlfWorld|BabyAI|$\Delta$ Held-in|$\Delta$ Held-out|
> |-|-|-|-|-|-|-|-|-|
> |Base model||28.59|31.19|33.63|26.56|67.00|-|-|
> |WebShop|SFT|77.46|23.63|10.13|1.00|32.51|+48.87|-22.78|
> ||GRPO|86.50|33.28|40.75|24.13|79.21|+57.91|+4.75|
> |TextCraft|SFT|10.99|23.84|70.25|2.69|62.67|+36.62|-13.29|
> ||GRPO|38.30|32.19|80.88|31.50|77.95|+47.25|+6.65|
>
> ### **# Question about failure analysis**
>
> Following your suggestions, we recruited 5 volunteers to perform error validation on 200 data (40 per environment). The consistency between human annotation and GPT annotation is 92.0%, demonstrating the reliability of our failure mode analysis. In cases of disagreement, annotators observed that GPT’s classification errors mainly occur in identifying "Confirmation Bias" and "State or Memory Inconsistency". In particular, GPT may overlook unverified erroneous information or repeated attempts in the trajectories. Full annotations will be added into the Appendix of our manuscript.
>
> ### **# Question about  influencing factors**
>
> Thank you for this insightful question. For the within-environment, cross-environment, and sequential training results we observed, we have provided several possible interpretations. Some are based on observations from our preliminary experiments and case studies, while others are informed by prior literature.
>
> **Actually, before conducting the large-scale experiments, we carried out a number of exploratory studies that provided useful priors for our experimental design.** For example, we examined different levels of validation strictness. In SearchQA, different evaluation schemes, such as partially exact match (sub-EM) and fully exact match (EM), led to substantially different generalization behaviors. We therefore adopted the setting that is more commonly used in in-domain practice. In addition, our preliminary findings suggest that the interaction mode, including the observation and action space, has a significant effect on cross-environment transfer. This was also one of the main reasons we conducted our experiments within a unified-interface framework.
>
> As you noted, additional ablation studies would make these conclusions more convincing. Due to the limited time and resources during the rebuttal period, we are unable to include more comprehensive ablations at this stage. We will incorporate these analyses in the next version.
>
> ---
>
> We once again thank you for your valuable comments! Your suggestions have provided us with a new perspective for explanation and analysis. If you have any further questions, please feel free to let us know and we will do our utmost to respond. If you find our replies satisfactory, we kindly ask that you consider updating your score accordingly.

---

> > ### Author Rebuttal · Reviewer_cMcU · 2026-04-01
> >
> > My concerns have been resolved, and I keep my postive score

---

> > > ### Author Response · Authors · 2026-04-06
> > >
> > > We are glad to have addressed your concerns. Your constructive suggestions have been incredibly helpful in refining our paper. Thank you again for your valuable time and positive feedback. We will do our utmost to revise our manuscript.

---

### Decision · Program_Chairs · 2026-04-30

**Decision:**

Accept (regular)

**Comment:**

The reviewers recognized the systematic empirical study of reinforcement fine-tuning (RFT) generalization for LLM agents, structured around a three-axis framework: intra-environment difficulty transfer, inter-environment transfer, and sequential multi-environment training. They noted the practical significance of the findings, particularly the insights into how agent generalization is sensitive to shifts in semantic priors and action-space interfaces. The effectiveness of the approach was supported by extensive experiments across five environments, two model sizes, and detailed qualitative failure analyses. However, concerns were raised regarding the incremental technical novelty, the heavy reliance on the existing AgentGym framework, the lack of algorithmic diversity in the initial submission, and the need for more rigorous stability metrics and baseline comparisons (e.g., against SFT).

The rebuttal addressed the reviewers’ points comprehensively by offering additional experimental results, including comparisons with SFT and the Reinforce++ algorithm, which demonstrated that the observed trends are consistent across different reinforcement learning paradigms. The authors also provided standard deviation metrics to verify the stability of their results and conducted a human-validation study to confirm the reliability of their GPT-based failure analysis.

After the rebuttal, the paper received two accept scores, one weak accept score, and one weak reject score. In the subsequent discussion phases, Reviewers cMcU, XnU7, and ezBg found the rebuttal largely satisfactory, particularly recognizing the authors' effort to enhance the empirical rigor and clarify the positioning of the work relative to prior single-turn reasoning studies. However, Reviewer CHDi maintained reservations regarding the technical innovation, pointing out that the results could be perceived as a direct application of the AgentGym and AgentGym-RL frameworks. Conversely, Reviewers XnU7 and ezBg supported the paper's acceptance, highlighting that the core contribution lies in the novel empirical study design and the resulting insights into agent transferability, which are not automatically provided by existing benchmarks.

ACs have read the paper, reviews, rebuttal, and discussion. ACs found the arguments of the supporting reviewers to be the most persuasive and weighted the paper's empirical strengths over the concerns about architectural novelty. While the underlying frameworks are pre-existing, the paper demonstrates that a systematic characterization of RFT generalization in multi-turn agents provides new and actionable insights for real-world deployment. Notably, the paper introduces a rigorous analysis of failure modes and establishes a connection between cross-environment and sequential-training generalization. These contributions provide the community with a valuable baseline and interesting insights for future research into autonomous agent scaling.

ACs reached a consensus on acceptance. The authors are encouraged to improve the final paper version by following reviewer recommendations, especially by moving the limitations section to the main paper, integrating the stability and hyperparameter ablation results, and clearly articulating the distinction between multi-turn agent generalization and static single-turn reasoning tasks.